# Casein kinase 1 dynamics underlie substrate selectivity and the PER2 circadian phosphoswitch

Jonathan M Philpott[1†], Rajesh Narasimamurthy[2†], Clarisse G Ricci[3†], Alfred M Freeberg[1], Sabrina R Hunt[1], Lauren E Yee[1], Rebecca S Pelofsky[1], Sarvind Tripathi[1], David M Virshup[2,4]*, Carrie L Partch[1,5]*

[1]Department of Chemistry and Biochemistry, University of California Santa Cruz, Santa Cruz, United States; [2]Program in Cancer and Stem Cell Biology, Duke-NUS Medical School, Singapore, Singapore; [3]Department of Chemistry and Biochemistry, University of California San Diego, San Diego, United States; [4]Department of Pediatrics, Duke University Medical Center, Durham, United States; [5]Center for Circadian Biology, University of California San Diego, San Diego, United States

**Abstract** Post-translational control of PERIOD stability by Casein Kinase 1$\delta$ and $\varepsilon$ (CK1) plays a key regulatory role in metazoan circadian rhythms. Despite the deep evolutionary conservation of CK1 in eukaryotes, little is known about its regulation and the factors that influence substrate selectivity on functionally antagonistic sites in PERIOD that directly control circadian period. Here we describe a molecular switch involving a highly conserved anion binding site in CK1. This switch controls conformation of the kinase activation loop and determines which sites on mammalian PER2 are preferentially phosphorylated, thereby directly regulating PER2 stability. Integrated experimental and computational studies shed light on the allosteric linkage between two anion binding sites that dynamically regulate kinase activity. We show that period-altering kinase mutations from humans to *Drosophila* differentially modulate this activation loop switch to elicit predictable changes in PER2 stability, providing a foundation to understand and further manipulate CK1 regulation of circadian rhythms.

*For correspondence:
david.virshup@duke-nus.edu.sg
(DMV);
cpartch@ucsc.edu (CLP)

†These authors contributed
equally to this work

Competing interests: The
authors declare that no
competing interests exist.

Reviewing editor: Yibing Shan,
DE Shaw Research, United States

## Introduction

Circadian rhythms are generated by a set of interlocked transcription/translation feedback loops that elicit daily oscillations in gene expression to confer temporal regulation to behavior, metabolism, DNA repair and more (*Bass and Lazar, 2016*). The PERIOD proteins (PER1 and PER2) nucleate assembly of large, multimeric complexes with the circadian repressors CRY1 and CRY2 that directly bind to and inhibit the core circadian transcription factor, CLOCK:BMAL1, on a daily basis (*Aryal et al., 2017*; *Michael et al., 2017*; *Xu et al., 2015*). PERs are stoichiometrically limiting for the assembly of these essential repressive complexes (*Lee et al., 2011b*). In this way, their abundance and post-translational modification state relay important biochemical information on the relative timing of the clock to other core clock proteins. Therefore, the expression, modification, and protein stability of PER1 and PER2 is under particularly tight regulation.

While both transcriptional and post-transcriptional mechanisms feature importantly in the rhythmic generation of PER proteins (*Kojima et al., 2011*; *Takahashi, 2017*), much attention has been focused on the post-translational control of PER orchestrated by its cognate kinases, CK1$\delta$ and the closely related paralog CK1$\varepsilon$ (*Hirano et al., 2016*). These clock-associated kinases are somewhat unusual, in that they remain stably anchored to PER1 and PER2 throughout the circadian cycle (*Aryal et al., 2017*; *Lee et al., 2001*) via a conserved Casein Kinase Binding Domain (CKBD)

(*Eide et al., 2005*; *Lee et al., 2004*). Mutations in CK1δ/ε (hereafter referred to jointly as CK1), as well as PER2, exert powerful control over circadian period, altering the intrinsic timing of circadian rhythms by hours in vivo (*Lowrey et al., 2000*; *Toh et al., 2001*; *Xu et al., 2005*; *Xu et al., 2007*). Because circadian period is linked to the timing of sleep onset, PER2 or CK1-dependent alterations to human circadian period manifest as sleep phase disorders that influence behavior and wellbeing on a daily basis (*Jones et al., 2013*).

PER2 is regulated by a CK1-dependent phosphoswitch, where kinase activity at two antagonistic sites functionally interact to control PER2 stability (*Masuda et al., 2019*; *Zhou et al., 2015*). Two features define the CK1 phosphoswitch: degradation is initiated by phosphorylation of a Degron located several hundred residues upstream of the CKBD to recruit the E3 ubiquitin ligase, β-TrCP (*Eide et al., 2005*; *Vanselow et al., 2006*); this is counteracted by sequential phosphorylation of five serines embedded within the CBKD known as the FASP region (*Narasimamurthy et al., 2018*). This region is named for a Ser to Gly polymorphism in human PER2 that disrupts this stabilizing multi-site phosphorylation, shortens circadian period, and leads to Familial Advanced Sleep Phase Syndrome (*Toh et al., 2001*). Mutation of the Degron phosphorylation site has the opposite effect, stabilizing PER2 to significantly compromise circadian rhythms (*Reischl et al., 2007*) in a manner similar to its constitutive overexpression (*Chen et al., 2009*). Therefore, the balance of stabilizing and degrading phosphorylation by CK1 leads to a complex temporal pattern of degradation in PER2 that is important for circadian timing (*Zhou et al., 2015*).

Despite the importance of CK1 for circadian timing in eukaryotic organisms from humans to *Drosophila*, *Neurospora*, and green algae (*Görl et al., 2001*; *Kloss et al., 1998*; *van Ooijen et al., 2013*; *Xu et al., 2005*), little is known about how its activity is regulated on clock protein substrates. CK1 is thought of as an anion- or phosphate-directed kinase, relying on negative charge on the substrate to template activity in the pSxxS consensus motif (*Flotow et al., 1990*). Consistent with this, biochemical studies exploring CK1 activity have primarily relied on the non-physiological acidic substrates casein (*Venerando et al., 2014*) and phosvitin (*Lowrey et al., 2000*), or used peptides harboring anion- or phosphate-driven motifs (*Isojima et al., 2009*; *Marin et al., 1994*; *Shinohara et al., 2017*). However, in vitro studies of clock-relevant kinase mutants using these non-physiological substrates have led to the puzzling conclusion that CK1 mutants that either decrease or increase period length all have reduced kinase activity (*Kivimäe et al., 2008*; *Venkatesan et al., 2019*).

The importance of anion binding is highlighted by the CK1ε *tau* allele that markedly speeds up the clock, leading to a ~ 20 hr period (*Lowrey et al., 2000*; *Ralph and Menaker, 1988*). The R178C mutation in *tau* alters a phosphate binding pocket on the surface of the kinase to block the ability of CK1 to further phosphorylate primed or acidic substrates. We discovered that *tau* has reduced activity on the FASP region in vitro, but exhibits a gain of function on the Degron site. Therefore, the *tau* allele inverts substrate selectivity on PER2 relative to the wild-type kinase to promote PER2 degradation. A mechanism for inverted substrate selectivity was suggested by the crystal structure of the *tau* kinase domain, which revealed the presence of a two-state conformational switch in the CK1 activation loop. Anion binding near the activation loop biases the switch towards a conformation that favors the FASP substrate, while mutations that disfavor anion binding enhance activity towards the Degron. Molecular dynamics simulations reveal that the alternate conformation is stabilized in *tau*, forming the basis for its enhanced activity on the Degron. A comprehensive analysis of other short period kinase mutants from *Drosophila* to humans finds that they differentially bias this intrinsic switch to enhance phosphorylation of the Degron and turnover of PER2. Therefore, the anion-triggered activation loop switch may be a general mechanism regulating CK1 substrate selection.

## Results

### The tau mutant has decreased activity on the FASP region

We recently demonstrated that CK1 primes FASP phosphorylation in a slow, rate-limiting step at serine 659 (mouse PER2 numbering, *Figure 1A–B*), with phosphorylation of the downstream consensus sites following rapidly in a sequential manner (dashed arrows at pSxxS consensus, *Figure 1B*) (*Narasimamurthy et al., 2018*). Because *tau* is a loss of function on the FASP domain, the prevailing model was that this disabled the phosphoswitch to allow for unchecked Degron phosphorylation to degrade PER2 (*Gallego et al., 2006*). The *tau* mutation eliminates a positively charged residue in

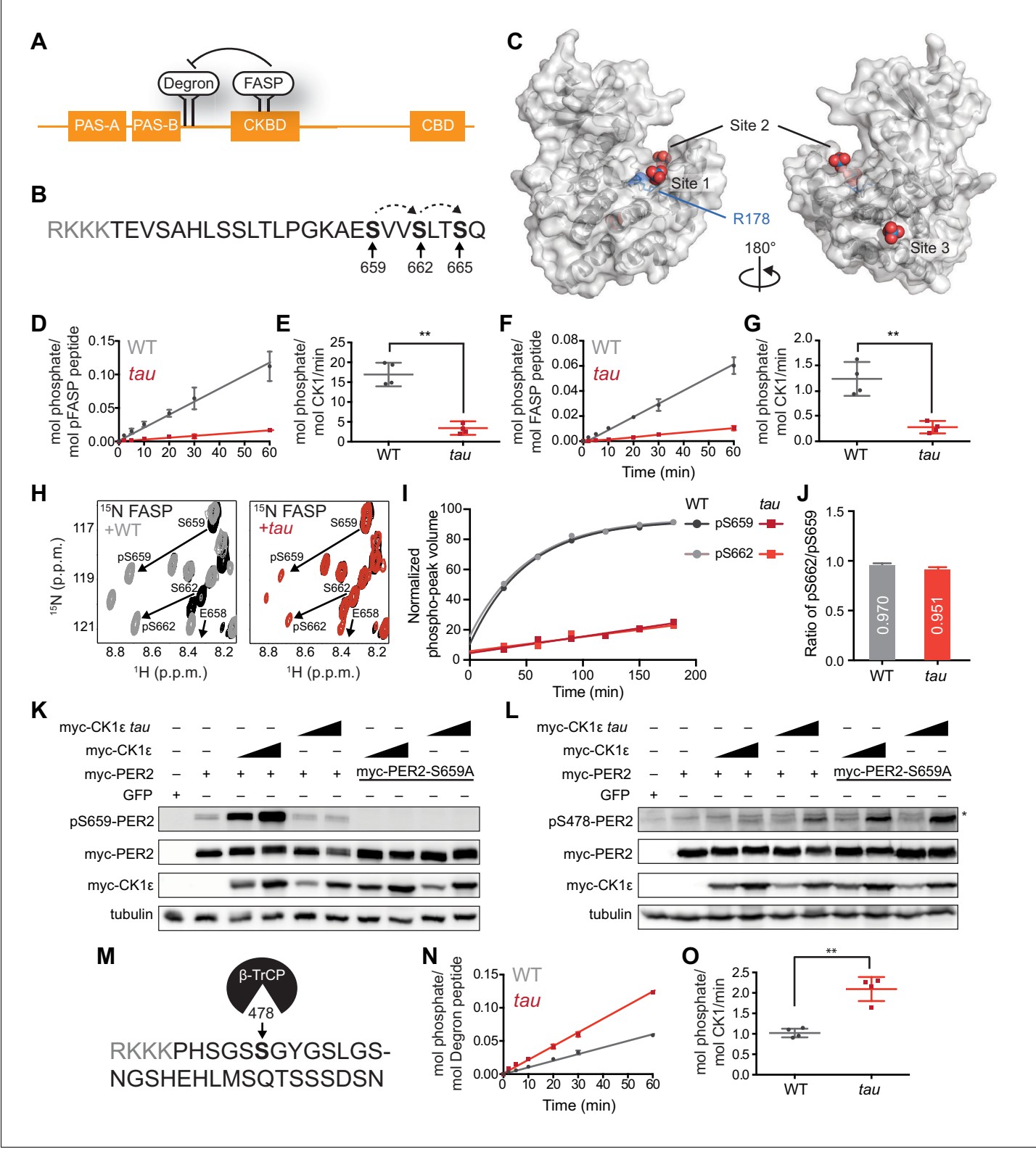

**Figure 1.** *tau* alters CK1 substrate selectivity on PER2 to enhance Degron phosphorylation. (**A**) Domain map of PER2 with tandem PAS domains, Casein Kinase-binding domain (CKBD), CRY-binding domain (CBD) and CK1 phosphorylation sites. (**B**) Sequence of the mouse PER2 FASP peptide with the priming site (S659, bold) and two downstream phosphorylation sites (S662 and S665, bold) that are phosphorylated sequentially by CK1δ (dashed arrows). Gray, polybasic motif included for $^{32}$P kinase assay. (**C**) CK1δ kinase domain with three anion binding sites (PDB: 1CKJ with $WO_4^{2-}$ anions). R178, blue. D, Kinase assay with 20 nM CK1δ ΔC WT or *tau* on 200 μM of primed FASP peptide (pS659) (n = 4 with s.d.). (**E**) Phosphorylation rates on

*Figure 1 continued on next page*

*Figure 1 continued*

primed FASP (n = 4 with s.d.). Significance assessed by unpaired Student's two-sided t-test: **, p<0.01. (F) Kinase assay as in D, but with 200 nM CK1δ ΔC WT or *tau* on 200 µM of unprimed FASP (n = 4 with s.d.). (G) Phosphorylation rates on the unprimed FASP peptide (n = 4 with s.d.). Significance assessed as above. (H) Overlaid $^{15}$N/$^{1}$H HSQC spectra at 3 hr timepoint in the NMR kinase assay on 200 µM $^{15}$N FASP (black) ±1 µM WT (gray) or *tau* (red) CK1δ ΔC. Arrows, phospho-specific peaks corresponding to pS659 and pS662. (I) Phosphoserine peak intensities for pS659 and pS662 by WT and *tau* kinases from NMR kinase assay. (J) Ratio of consensus to priming activity on the FASP (pS662/pS659) in the NMR kinase assay. Errors were estimated from the standard deviation of the noise in the spectrum. (K-L) Western blot of FASP priming site, detecting pS659 (K) or the Degron, detecting pS478 (L) on mouse myc-PER2 in HEK293 cell lysates after transfection with indicated expression plasmids. Representative blot from n = 3 shown. Wedge, 10 or 50 ng of myc-CK1ε plasmid used. *, non-specific band. (M) Sequence of mouse PER2 Degron peptide with S478 (bold) and polybasic motif (gray). (N) Kinase assay with 200 nM kinase on 200 µM Degron peptide (n = 4 with s.d.). (O) Phosphorylation rates on Degron (n = 4 with s.d.). Significance assessed as above. See also *Figure 1—figure supplement 1*.

The online version of this article includes the following figure supplement(s) for figure 1:

**Figure supplement 1.** The *tau* mutation alters substrate selectivity.

the first of three CK1 family-specific anion-binding sites (Sites 1, 2, and 3) (*Figure 1C*) (*Longenecker et al., 1996*). R178 sits at Site 1, located adjacent to the active site, and has been postulated to bind a phosphorylated priming site to position the downstream serine of the consensus motif near the active site (*Longenecker et al., 1996*; *Zeringo and Bellizzi, 2014*). Based on this model, we predicted that *tau* should preferentially disrupt phosphorylation of the downstream consensus sites in the FASP region due to its inability to recruit the primed substrate.

To test this idea, we used a FASP peptide based on the native mouse PER2 sequence (*Figure 1B*) that was primed synthetically by phosphorylation at S659. We used a constitutively active version of the isolated wild-type (WT) or *tau* (R178C) kinase lacking its autoinhibitory tail (CK1δ ΔC, with 97% identity between CK1δ and CK1ε in the kinase domain). As expected, *tau* had significantly lower activity than the WT kinase on this primed substrate (*Figure 1D–E*). This was also true for a minimal, primed synthetic substrate CK1tide (*Figure 1—figure supplement 1*). We then asked if *tau* would influence priming phosphorylation using an unmodified FASP peptide. As we observed before, phosphorylation of the non-consensus priming site occurs with much slower kinetics than the downstream consensus sites (*Figure 1F–G*) (*Narasimamurthy et al., 2018*). To our surprise, we found that *tau* also had significantly diminished activity on an unprimed FASP substrate (*Figure 1F–G*), indicating that R178 is also important for the non-consensus priming event. The decrease in activity of *tau* on the priming site was also validated using an ELISA-based kinase assay with an antibody that is specific for phosphorylated S659 (*Figure 1—figure supplement 1*).

We used an NMR-based kinase assay to confirm that *tau* influences both priming and downstream events at the FASP region. In contrast to radiolabeled kinase assays, this assay provides site-specific information on the substrate by measuring new peaks that arise for phosphorylated serines over time (*Theillet et al., 2013*). We established the requirement for priming to initiate phosphorylation of downstream sites in the FASP by CK1 (*Narasimamurthy et al., 2018*). Therefore, if *tau* was simply deficient in recruitment of primed substrate, we should observe a similar degree of phosphorylation at the priming site (pS659) compared to WT kinase, but a decreased peak volume for the downstream serine (pS662). By contrast, we observed that the peaks for both pS659 and pS662 were decreased in volume for *tau* (*Figure 1H–I*), revealing that the ratio of consensus to priming activity (pS662/pS659) was similar in WT CK1 and *tau* (*Figure 1J*).

To see if these findings held in the context of full-length protein, we expressed myc-PER2 and WT or *tau* myc-CK1ε in HEK293 cells and assessed phosphorylation using an antibody specific for phosphorylation on the priming site at S659. Consistent with our in vitro studies, the activity of *tau* was much lower on the priming site relative to the WT kinase (*Figure 1K*). As expected, phosphorylation of the subsequent serine was also decreased with *tau* (*Figure 1—figure supplement 1*). To examine the possibility that loss of activity on the FASP region was specifically due to the cysteine mutation (R178C), we also tested an R178A mutant in the cell-based assay and found that it also exhibited much lower activity on the FASP priming site in a cell-based assay (*Figure 1—figure supplement 1*). Collectively, these data show that loss of R178 at the site one anion-binding pocket in *tau* leads to a reduction in both the slow priming step as well as the downstream sequential, primed phosphorylation of the FASP region.

## *Tau* exhibits a gain of function on the Degron

Using an antibody specific for phosphorylation of the CK1-dependent β-TrCP recruitment site at S478, we recapitulated the increased activity of *tau* on the Degron observed previously on myc-PER2 that was transiently expressed with the kinase in HEK293 cells (*Figure 1L*) (*Gallego et al., 2006*). Consistent with the phosphoswitch model that FASP phosphorylation antagonizes CK1 activity on the Degron (*Figure 1A*), we observed an increase in activity of WT kinase on the Degron with the S659A mutant that eliminates all phosphorylation of the FASP region (*Figure 1L*) (*Narasimamurthy et al., 2018*). We found that *tau* also had increased activity on the Degron in the S659A mutant, demonstrating that the *tau* mutant can still be regulated by the phosphoswitch. Given that kinase activity on the Degron is clearly linked to phosphorylation of the FASP region in cells, we sought to clarify whether *tau* truly exhibits increased activity at the Degron using a peptide-based kinase assay in vitro (*Figure 1M*). Here, we found that even on a peptide substrate in vitro, *tau* had significantly increased activity on the Degron relative to WT kinase (*Figure 1N–O*). Moreover, both *tau* and WT kinase maintain their distinct substrate preference in the presence of both FASP and Degron peptides in vitro, with *tau* showing enhanced activity for the Degron relative to WT kinase (*Figure 1—figure supplement 1*). These data demonstrate that in addition to any regulation imparted by the phospho-FASP on the Degron in cells, the *tau* mutation leads to a fundamental change in CK1 activity and substrate specificity.

## The *tau* mutation disrupts anion binding at site one and site two on CK1

To explore the molecular basis for *tau's* altered substrate specificity, we solved a crystal structure of the CK1δ R178C kinase domain (*Figure 2A* and *Supplementary file 1a*). Both WT and *tau* coordinate an anion at Site three similarly (*Figure 2—figure supplement 1*). However, the mutation disrupted anion binding at Site 1, although it led to only minor structural changes in this anion-binding pocket (*Figure 2B*). Unexpectedly, we observed a loss of anion binding at Site two in *tau*, mediated by an alternate conformation of the activation loop near this pocket that sterically blocks binding of the anion (*Figure 2C*). This alternate conformation initiates at G175, three residues upstream of the *tau* mutation (*Figure 2C*). A rotation of the backbone at G175 to a left-handed configuration dramatically alters the configuration of upstream residues to create a distinct conformation of the activation loop (*Figure 2D and* ). A backbone flip of a glycine at this conserved position has been observed in other serine/threonine kinases, linking changes in conformation of the activation loop to regulation of kinase activity (*Nolen et al., 2004*); therefore, the 'loop up' conformation observed in *tau* may lead to different kinase activity than the 'loop down' conformation observed in the WT kinase.

The activation loop is a key feature that distinguishes the CK1 family from other serine/threonine kinases. There is little mechanistic insight into CK1 substrate selectivity because it deviates from the highly conserved APE motif in the P+1 region that helps to define substrate specificity in most kinases (*Figure 2E*) (*Goldsmith et al., 2007*). The activation loop and surrounding region also contain all of the residues that coordinate the three anions observed in nearly all CK1 structures: the residues that coordinate binding at Sites 1 and 3 are unique to the CK1 family, while R127 and K154, corresponding to Site 2, are broadly conserved in other kinase families. R127 is part of the highly conserved HRD motif that coordinates a phosphorylated residue in the activation loop of many kinases to regulate substrate binding and activity (*Figure 2F*) (*Johnson et al., 1996*). CK1 family kinases are generally considered to be constitutively active because they do not require phosphorylation of the activation loop (*Goldsmith et al., 2007*). However, CK1δ and CK1ε are inhibited by autophosphorylation of their disordered C-terminal tails (*Graves and Roach, 1995*; *Rivers et al., 1998*) and potentially by the phosphorylated FASP region (*Figure 1L*). Therefore, these anion binding sites could represent the basis for a CK1-specific regulatory mechanism by facilitating the binding of phosphorylated C-terminal tails or substrates, and/or anionic signaling molecules (*Fustin et al., 2018*; *Kawakami et al., 2008*).

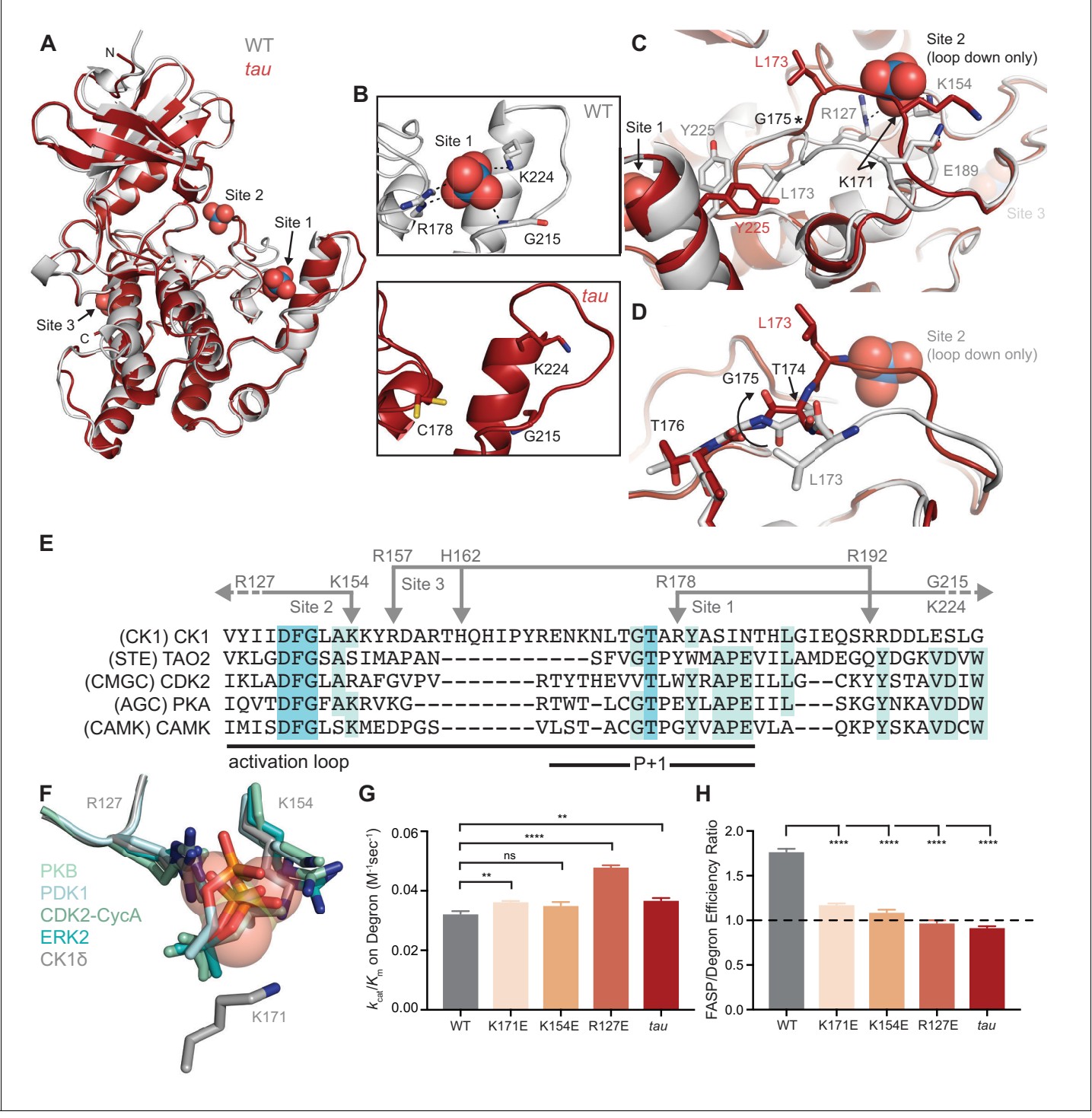

**Figure 2.** *tau* disrupts anion binding on CK1δ. (**A**) Overlay of WT kinase domain (gray, PDB: 1CKJ, chain B) with *tau* (maroon, PDB: 6PXN, chain A). The 3 anion binding sites (WO$_4^{2-}$, from 1CKJ) are labeled. (**B**) View of Site 1 in WT (top, gray) and *tau* (bottom, maroon). Polar interactions, dashed black lines. (**C**) Overlaid view of Site two in WT and *tau* as above. Polar interactions, dashed black lines. Asterisk, hinge point for conformational change at G175. Note: Site two anion is only bound in WT, as it is blocked by the activation loop in *tau*. (**D**) Representation depicting the left-hand configuration of G175 and subsequent rotation (solid arrow) of upstream residues T174 and L173. (**E**) Alignment of the activation loop of CK1δ with representatives of other serine/threonine kinase families. Residues that coordinate anion binding on CK1 are indicated above in gray. (**F**) Superposition of the Site two anion binding site in CK1 with the binding site for the phosphorylated activation loop of other serine/threonine kinases. Depicted are: PKB (PDB: 1O6K, pale cyan), PDK1 (1H1W, aquamarine), CDK2 (1QMZ, green cyan), ERK2 (2ERK, teal) and CK1δ (5 × 17, dark gray). Residues that coordinate the anion are depicted in sticks, as are phosphoserine or phosphothreonine residues from other kinases; the SO$_4^{2-}$ coordinated at Site two by CK1δ (PDB: 5 × 17)

*Figure 2 continued on next page*

*Figure 2 continued*

is shown in transparent spheres. (**G**) Enzymatic efficiency on the Degron (n = 3 with s.d.). Significance assessed relative to WT with an unpaired Student's two-sided t-test: **, p<0.01; ***, p<0.001; ****, p<0.0001. (**H**) Ratio of enzymatic efficiency on FASP relative to Degron (n = 3 with s.d.). Equivalent activity on FASP and Degron, dashed line. Significance assessed as above. See also *Figure 2—figure supplement 1*.

The online version of this article includes the following figure supplement(s) for figure 2:

**Figure supplement 1.** Details of CK1 crystal structures.

## Eliminating anion binding at site 2 differentially regulates CK1 activity on the FASP and degron

Mutation of the residues corresponding to positions R127 and K154 at Site 2 essentially eliminates the activity of kinases that depend on phosphorylation of the activation loop (*Gibbs and Zoller, 1991*; *Leon et al., 2001*; *Skamnaki et al., 1999*). To test the role of Site 2 anion binding in regulation of CK1 activity, we made charge reversion mutants at positions R127 and K154, as well as at K171 located nearby on the activation loop, and measured enzymatic efficiency ($k_{cat}/K_m$) on FASP and Degron peptides in vitro. We observed a modest decrease (~25%) in activity towards the FASP peptide (*Figure 2—figure supplement 1* and *Supplementary file 1b*), suggesting that the 'loop down' conformation that is enforced by anion binding at Site 2 may be important for FASP activity. Strikingly, these mutants all increased activity towards the Degron, with a ~ 50% gain in efficiency for R127E (*Figure 2—figure supplement 1* and *Figure 2G*). To further examine the role of K171 in regulation of anion binding, we solved a structure of the K171E mutant crystallized in high sulfate conditions and found a full complement of three anions bound (*Figure 2—figure supplement 1*), demonstrating that local flexibility in the activation loop allows it to retain sulfate binding at Site 2 to some degree. Overall, these data suggest that anion binding at Site 2 correlates with substrate selectivity on the FASP and Degron peptides. Notably, the change in selectivity with Site 2 mutants in vitro makes them much more *tau*-like (*Figure 2H*), consistent with a recent report that both *tau* and charge reversion mutants at Site 2 lead to decreased PER2 stability in cells (*Shinohara et al., 2017*).

## The activation loop switch is intrinsic to the CK1 family of kinases

Two copies of the kinase are found in the asymmetric unit of the *tau* crystal. We discovered that the mutant kinase can take on either the 'loop down' or the 'loop up' conformation (*Figure 3A*). The activation loop is not stabilized by crystal contacts in the alternate 'loop up' conformation and was explicitly modeled based on good density in both conformations (*Figure 3—figure supplement 1*), suggesting that the kinase has an intrinsic ability to take on two discrete conformations in a switch-like manner. Since the first structure of CK1δ published in 1996, nearly all CK1 structures have been determined after crystallization with high concentrations of sulfate or citrate anions (*Long et al., 2012*; *Longenecker et al., 1996*; *Minzel et al., 2018*). Because anion binding at Site two is incompatible with the activation loop in its 'loop up' conformation, prior crystallographic conditions have likely disfavored this alternate conformation. The WT structure that we used for our analysis (PDB: 1CKJ) was first crystallized with a low concentration of anions, and then derivatized with tungstate as an analog for phosphate before data collection (*Longenecker et al., 1996*). Importantly, this structure also displays the same two discrete conformations marked by translocation of residue L173 in the activation loop (*Figure 3A*), confirming that this switch is an intrinsic property of the CK1δ kinase that has not been explored functionally.

We probed the library of existing CK1δ structures using the positioning of L173 as a quantitative metric for activation loop conformation by measuring interatomic distances of the L173 CD2 atom to either CD2 of L152 (short distance in the 'loop up' conformation, long in the 'loop down') or the hydroxyl of Y225 (long in the 'loop up', short in the 'loop down') (*Figure 3B*). A survey of 68 chains from 26 different crystal structures of CK1δ (in at least 7 space groups) demonstrated that *tau* and the WT kinase from PDB entry 1CKJ are the only structures of CK1δ to have ever been captured in the 'loop up' conformation (*Figure 3C*). Moreover, the residues that coordinate anion binding are broadly conserved in the CK1 family and anions are also found bound in these sites in other CK1 family kinases (e.g., CK1ε, CK1γ3; *Supplementary file 1c*). To determine if we could independently capture the two states of the activation loop switch in the native kinase, we optimized sulfate-free

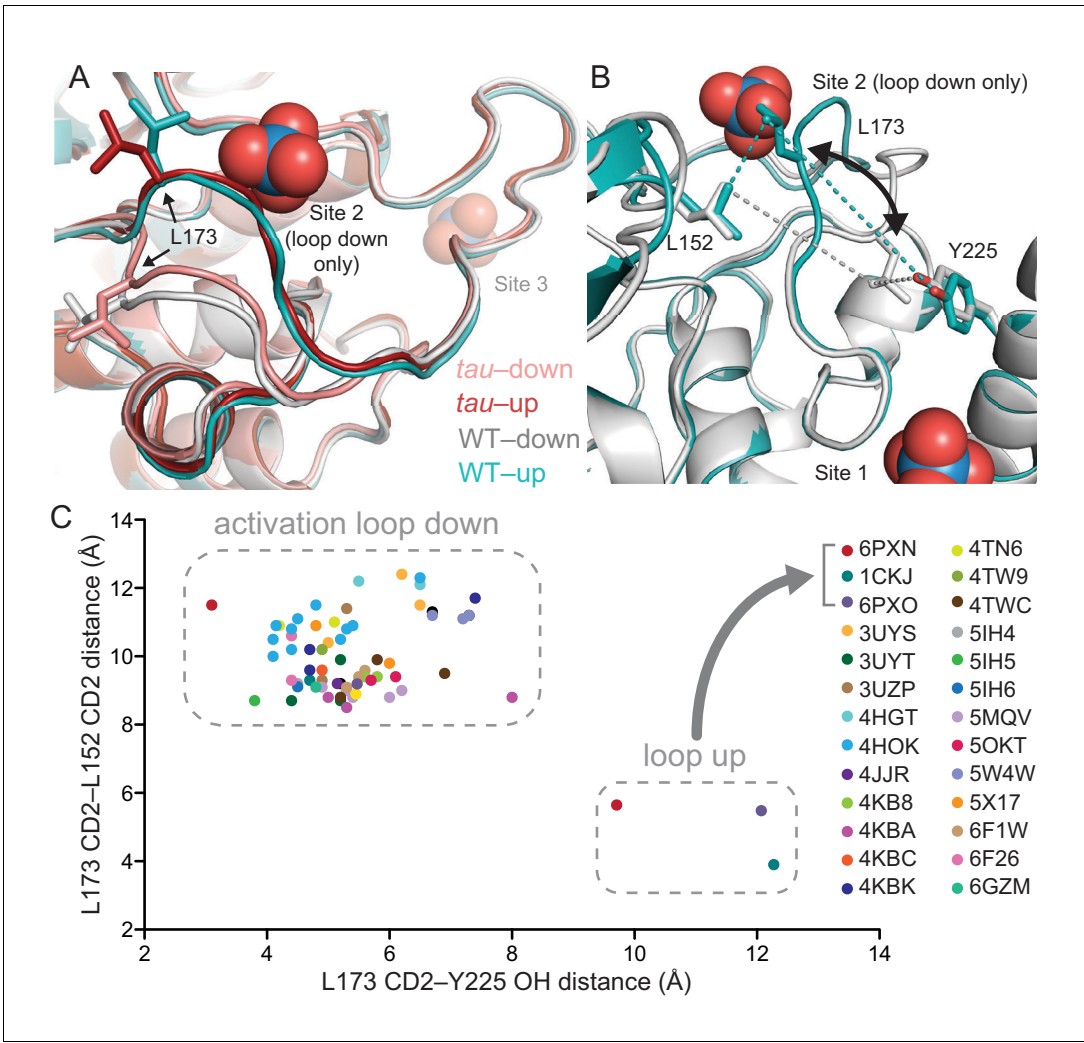

**Figure 3.** *tau* alters an intrinsic molecular switch in the activation loop of CK1δ. (**A**) View of the activation loop switch in WT (PDB: 1CKJ, chains A (cyan) and B (gray)) and *tau* (PDB: 6PXN, chains A (maroon) and B (salmon)). (**B**) Position of L173 CD2 relative to either L152 CD2 or Y225 OH, reporting on the conformation of the activation loop in the 'loop down' (gray) or 'loop up' (cyan) conformation in WT CK1δ (PDB: 1CKJ). (**C**) Scatter plot of interatomic distances in Å (from panel B) measured in 68 chains from 26 different crystal structures of CK1δ ΔC. See *Figure 3—figure supplement 1* and *Supplementary file 1c* for more information.

The online version of this article includes the following figure supplement(s) for figure 3:

**Figure supplement 1.** Crystallographic details of the CK1 activation loop switch.

crystallographic conditions and solved a structure of anion-free CK1δ. Similar to the 1CKJ structure, we observed both conformations of the activation loop in the two molecules of the asymmetric unit (*Figure 3—figure supplement 1*). Therefore, CK1 has an anion-dependent switch in its activation loop. However, the *tau* mutation appears to favor the 'loop up' conformation because it was observed in crystals that grew in the presence of high sulfate concentrations. Taken together, these data suggest that the *tau* mutation allosterically regulates anion binding at Site 2 via the activation loop.

### *Tau* stabilizes the rare 'loop up' conformation of the CK1 activation loop

To probe the dynamic behavior of CK1 and how this is perturbed by the *tau* mutation, we performed Gaussian Accelerated Molecular Dynamics (GaMD) simulations (*Miao et al., 2015*) on four systems: WT and *tau* CK1 with the activation loop in the crystallographically-defined 'up' or 'down'

conformations (*Supplementary file 1d*). By monitoring the Root Mean Square Deviation (RMSD) of the activation loop with respect to the 'down' or 'up' crystallographic conformations, we set out to assess its stability over the course of 500 ns simulations (*Figure 4A–D*). We found that the activation loop remained stably in position when simulations were started from the 'loop down' conformation for both *tau* and WT. Similar results were seen when the anion was computationally removed from Site 2, suggesting that this conformation of the activation loop is intrinsically stable (*Figure 4—figure supplement 1*). However, in simulations starting from the 'loop up' conformation, the WT activation loop rapidly underwent a conformational change, as shown by increased $RMSD^{up}$ values. Because we did not see a concomitant decrease in the $RMSD_{down}$ values, we can conclude that this is not a complete transition from 'loop up' to 'loop down' on this timescale. Importantly, these transitions occurred more frequently in WT CK1 than in *tau* (*Figure 4C–D*). This confirms that the 'loop up' conformation is better tolerated in *tau*, consistent with our observation of this apparently rare conformation in our crystal structure. We also observed that Y225 displayed more conformational freedom in *tau* compared to the WT kinase (*Figure 4—figure supplement 1*). Given that Y225 is directly adjacent to K224 in Site one and it also makes contact with L173 in the activation loop in its 'loop down' conformation, this suggests that Y225 could be poised to coordinate the conformation of the activation loop with activity at Site 1.

## The activation loop allosterically controls the dynamics of loop L-EF in *tau*

While monitoring the overall dynamics of WT and *tau* CK1, we detected a major difference in the dynamics of the loop connecting α-helices E and F (loop L-EF, *Figure 4—figure supplement 1*), part of the anion binding site disrupted by the *tau* mutation (*Figure 4E–H*). This is intriguing because temperature-dependent dynamics of loop L-EF were recently shown to be important for the temperature-compensated activity of CK1 on PER2 (*Shinohara et al., 2017*). In our simulations of the WT kinase, this loop exhibited relatively restricted mobility, regardless of whether the activation loop was in the 'up' or 'down' conformation (*Figure 4E and G*). Similar results were observed when the anion was computationally removed from Site 2 (*Figure 4—figure supplement 1*). However, we observed that the 'loop down' conformation of the activation loop was accompanied by a strong disorganization of loop L-EF in *tau* (*Figure 4H*), leading to significantly larger conformational freedom compared to the WT kinase (*Figure 4G*). Surprisingly, the enhanced dynamics of loop L-EF was not observed when *tau* was in the 'loop up' conformation; instead, loop L-EF displayed the same restricted dynamics as the WT kinase (*Figure 4E-F*. These data suggest that the disruption of Site one by loop L-EF dynamics in the *tau* mutation may be due to allosteric communication with the activation loop, and by proxy, anion binding at Site 2.

## *Tau* dynamically reshapes the substrate-binding cleft in CK1

To investigate how the activation loop and dynamics of loop L-EF might influence substrate selectivity, we calculated the volume of the substrate binding cleft and adjacent anion binding sites throughout the 500 ns GaMD simulations (*Figure 4* and *Figure 4—figure supplement 1*). As expected, Site two was completely open only when the activation loop was in the 'loop down' conformation for the WT kinase (*Figure 4I*). Interestingly, this site became partially open in simulations of the WT kinase starting from the 'loop up' conformation (*Figure 4J*), indicating that the intermediate conformational state observed in our simulations might allow the kinase to recover, to some extent, the ability to bind an anion at Site 2. The conformation of the activation loop indirectly affects the volume of the substrate binding cleft with opposing effects in WT and *tau*. In WT, the substrate binding cleft was open more consistently with the activation loop in its preferred 'down' conformation (*Figure 4I–J*). By contrast, the substrate binding cleft was open more consistently in *tau* in the 'loop up' conformation (*Figure 4L*) and, due to the dynamic disordering of loop L-EF, often closed when in the 'loop down' conformation (*Figure 4K*). Because the activation loop does not contact the substrate binding cleft, it cannot directly affect the shape of the cleft by steric effects. Instead, closing of the substrate binding cleft in *tau* occurs due to the conformational disorganization in loop L-EF, which is allosterically induced when the activation loop is in the 'down' conformation.

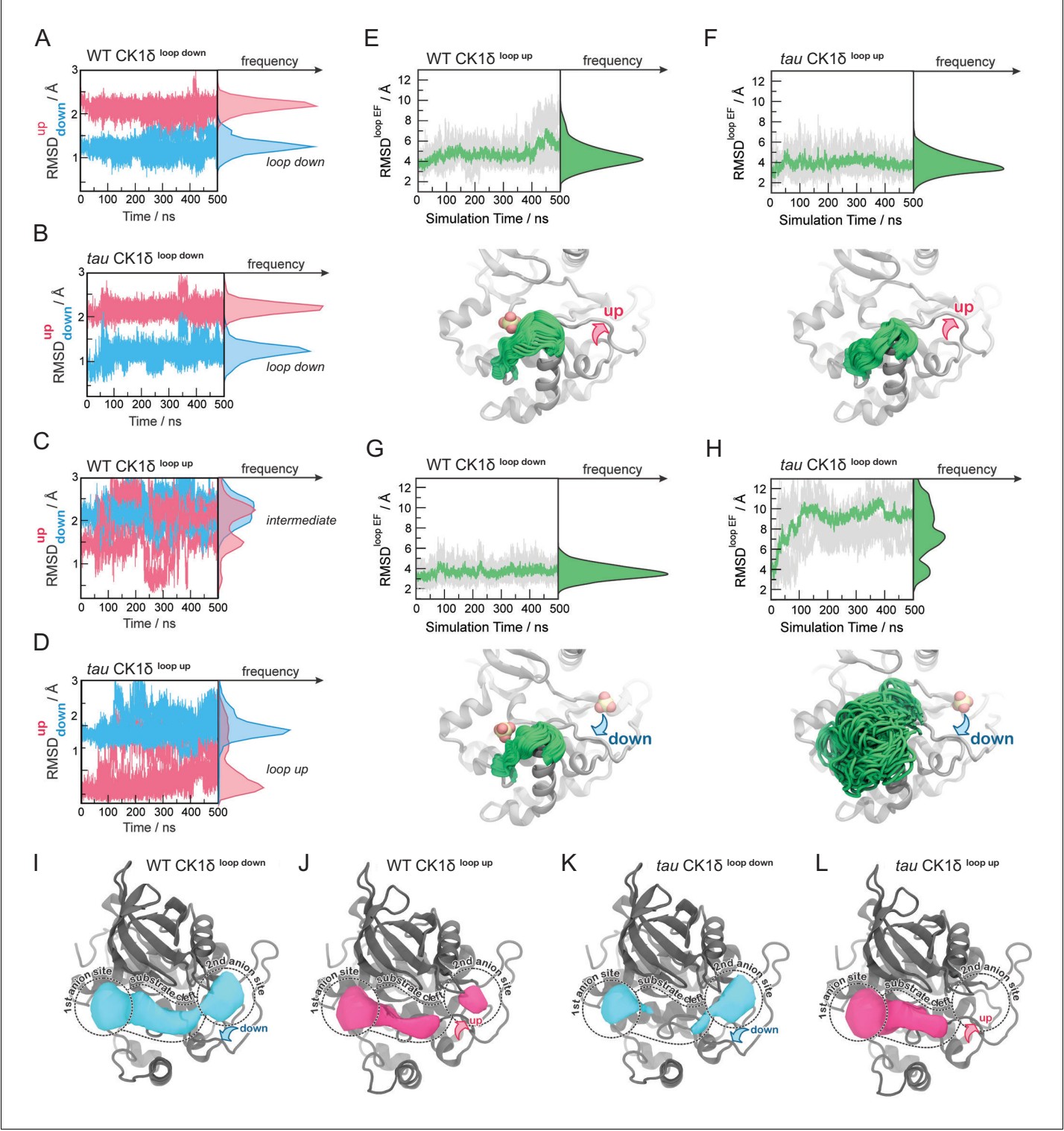

**Figure 4.** Probing the dynamics of CK1δ with GaMD simulations. (**A-D**) Stability of the activation loop assessed by the RMSD of residues 168–175 with respect to the 'loop down' (RMSD$_{down}$, blue) or 'loop up' conformation (RMSD$^{up}$, pink), as observed in the crystal structure. For each system, the RMSDs from all five MD replicas are superimposed. Panel A, WT CK1δ$^{loop\ down}$; (**B**) tau CK1δ$^{loop\ down}$; C, WT CK1δ$^{loop\ up}$; D, tau CK1δ$^{loop\ up}$. (**E-H**) Dynamics of the EF loop assessed by the RMSD of residues 213–224 with respect to the initial structure. For each system, the RMSD was calculated for individual replica (gray lines, n = 5) and then averaged (green). The molecular representations in panels E-H show the crystallographic structure of CK1δ (gray) superimposed with snapshots of the L-EF loop extracted from the GaMD simulations (green). When present in the crystal structure and the simulation, sulfate anions are represented by spheres. I-L, Alterations in anion and substrate-binding clefts arise from the activation loop switch and tau

*Figure 4 continued on next page*

*Figure 4 continued*

mutation. Volumes for the binding clefts were extracted and averaged from GaMD simulations in the four states: panel I, WT CK1δ^loop down^; J, WT CK1δ^loop up^; K, *tau* CK1δ^loop down^; L, *tau* CK1δ^loop up^. Water and anions were removed from the analysis. Volumetric maps are contoured at 0.1 and represent regions that were consistently open during the simulations. See also *Figure 4—figure supplement 1* and *Supplementary file 1d* for more information.

The online version of this article includes the following figure supplement(s) for figure 4:

**Figure supplement 1.** Additional details for GaMD simulations.

## *Tau* influences the global dynamics of CK1

We next used principal component analysis (PCA) to uncover effects of *tau* on the principal modes of motion displayed by CK1 during the GaMD simulations (*Figure 5—figure supplement 1* and *Video 1*). The 1st principal component consisted of a clear 'open-and-close' movement of the enzyme, achieved mainly by dislocation of the N-terminal lobe (N-lobe) with respect to the top of the helix F (*Figure 5A*). This mode of motion has been shown to control accessibility to the ATP-binding site and regulate substrate access in other kinases (*McClendon et al., 2014*). The histograms of the 1st principal component show that WT CK1 sampled more of the open conformations compared to *tau* (*Figure 5C*). The 2nd principal component consisted of a twisting movement of the N-lobe with respect to the top of helix F and a significant rearrangement of loop L-EF, which can either be extended or collapsed (*Figure 5B*). When collapsed, loop L-EF had the effect of sterically closing the substrate binding cleft. The histograms of the 2nd PCA illustrate that loop L-EF adopts more extended conformations in WT CK1 and more collapsed conformations in *tau* (*Figure 5D*). The collapsed conformations require loop L-EF to undergo a significant conformational change, which agrees with the high conformational freedom seen in this region when the *tau* activation loop is in the 'loop down' conformation. Altogether, the GaMD simulations suggest that the *tau* kinase never behaves fully like the WT enzyme; compared to WT, it stabilizes the rare, Degron-preferring conformation of the activation loop that remodels the substrate binding cleft and excludes anion binding at Site 2. By contrast, when *tau* samples the 'loop down' conformation of the activation loop, it leads to a dynamic disordering of loop L-EF that favors conformations of the kinase that likely accounts for its decrease enzymatic efficiency on the FASP region.

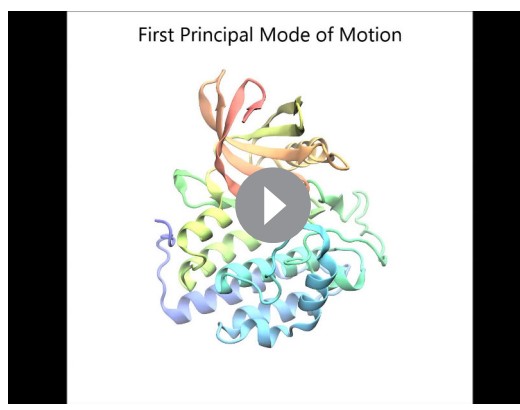

**Video 1.** Principal Component Analysis of CK1δ normal modes. The 1st principal mode of motion corresponds to an 'open-and-close' movement of the kinase, achieved mainly by dislocation of the N-terminal lobe (N-lobe) with respect to the top of the helix F. The 2nd principal mode of motion corresponds to a twisting movement of the N-lobe with respect to the top of helix F and significant rearrangement of loop L-EF, which can either be extended or collapsed.
https://elifesciences.org/articles/52343#video1

## Circadian alleles from *Drosophila* to humans occur throughout CK1

To gain further insight into CK1 function, we mapped known mutant alleles that influence circadian rhythms onto the CK1 structure (*Figure 6* and Supplemenary File 2). *Doubletime* (DBT), the CK1δ/ε ortholog in *Drosophila*, has one allele that causes a short circadian period (*dbt^S^*, P47S) while all others lead to a long period (*Kloss et al., 1998*; *Rothenfluh et al., 2000*; *Suri et al., 2000*). Many long period mutations occur at or near catalytically important residues like the catalytic DFG motif and the regulatory spine that controls kinase activity (*Figure 6B*) (*Taylor and Kornev, 2011*; *Muskus et al., 2007*). Moreover, two loss of function alleles occur in the activation loop: *dco^18^* (S181F) is located right behind the *tau* site (R178), linking it to the substrate binding channel and activation loop, while *dco^2^* (G175S) occurs at the hinge point for the activation loop switch (*Zilian et al., 1999*). Although *Drosophila* and mammalian PER proteins are somewhat functionally divergent, conservation of their Degrons (*Chiu et al.,*

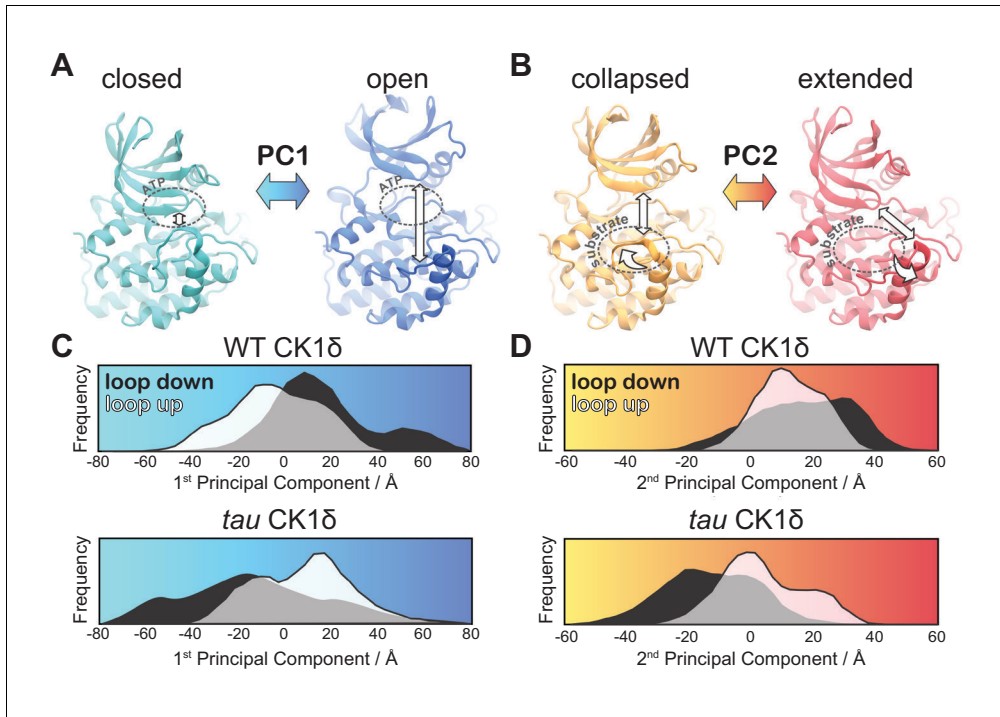

**Figure 5.** *tau* influences the principal modes of motion in CK1δ. (**A**) PC1 consists of an 'open-and-close' movement achieved by dislocation of the N-lobe with respect to the top of the helix F to control accessibility of the ATP-binding site (dotted circle). (**B**) PC2 consists of a twisting movement of N-lobe with respect to the top of helix F and significant rearrangement of the loop L-EF, which can be extended or collapsed against the substrate binding site (dotted circle). (**C-D**) Structural representations correspond to (panel A) PC1 = −80 Å (cyan), PC1 = 80 Å (blue) and (panel B) PC2 = −60 Å (orange), and PC2 = 60 Å (red). The histograms represent projections of the accumulated GaMD trajectories along the 1st (**C**) or 2nd (**D**) principal components for WT CK1δ and the *tau* mutant, either in the activation 'loop down' (black) or 'loop up' (white) conformations. See also *Figure 5—figure supplement 1* and *Video 1*.

The online version of this article includes the following figure supplement(s) for figure 5:

**Figure supplement 1.** Details of the principal modes of motion.

---

2008; *Eide et al., 2005*) and FASP-like stabilizing phosphorylation sites (*Kivimäe et al., 2008*; *Top et al., 2018*; *Xu et al., 2007*) suggests that there may be some conservation in their regulation by CK1 (*Figure 6C*). In line with this, expressing mammalian CK1δ with the *tau* or *dbt^s* mutation leads to short period circadian rhythms in flies (*Fan et al., 2009*). Consistent with this functional conservation, the entire surface-exposed area linking Sites 1 and 2 and the substrate binding cleft are ≥95% identical in 20 species from humans to unicellular green alga where CK1 has been implicated in regulation of circadian clocks (*Figure 6—figure supplement 1*).

## Other short period mutants exhibit differential activity on the FASP and degron

Three short period mutants from humans and *Drosophila* (T44A, H46R, and P47S) colocalize in the N-lobe right above Site 2 (*Figure 6B*) (*Kloss et al., 1998*; *Brennan et al., 2013*; *Xu et al., 2005*). Given the changes in N- to C-lobe dynamics that we observed in our simulations of CK1 (*Figure 5*), we wondered how these mutations would influence substrate selectivity. We first tested the activity of these short period kinase mutants in cell-based transfection assays by monitoring both FASP priming (*Figure 7A*) and Degron (*Figure 7B*) phosphorylation. Similar to the Site two mutants we tested earlier (*Figure 2—figure supplement 1*), the short period mutants T44A, H46R, and P47S each retained substantially more kinase activity on the FASP priming site than *tau*. A triple mutant of all three short period mutants (3M) did not have additive effects. Likewise, they also appeared to retain kinase activity or even exhibited modest increases in Degron phosphorylation relative to WT

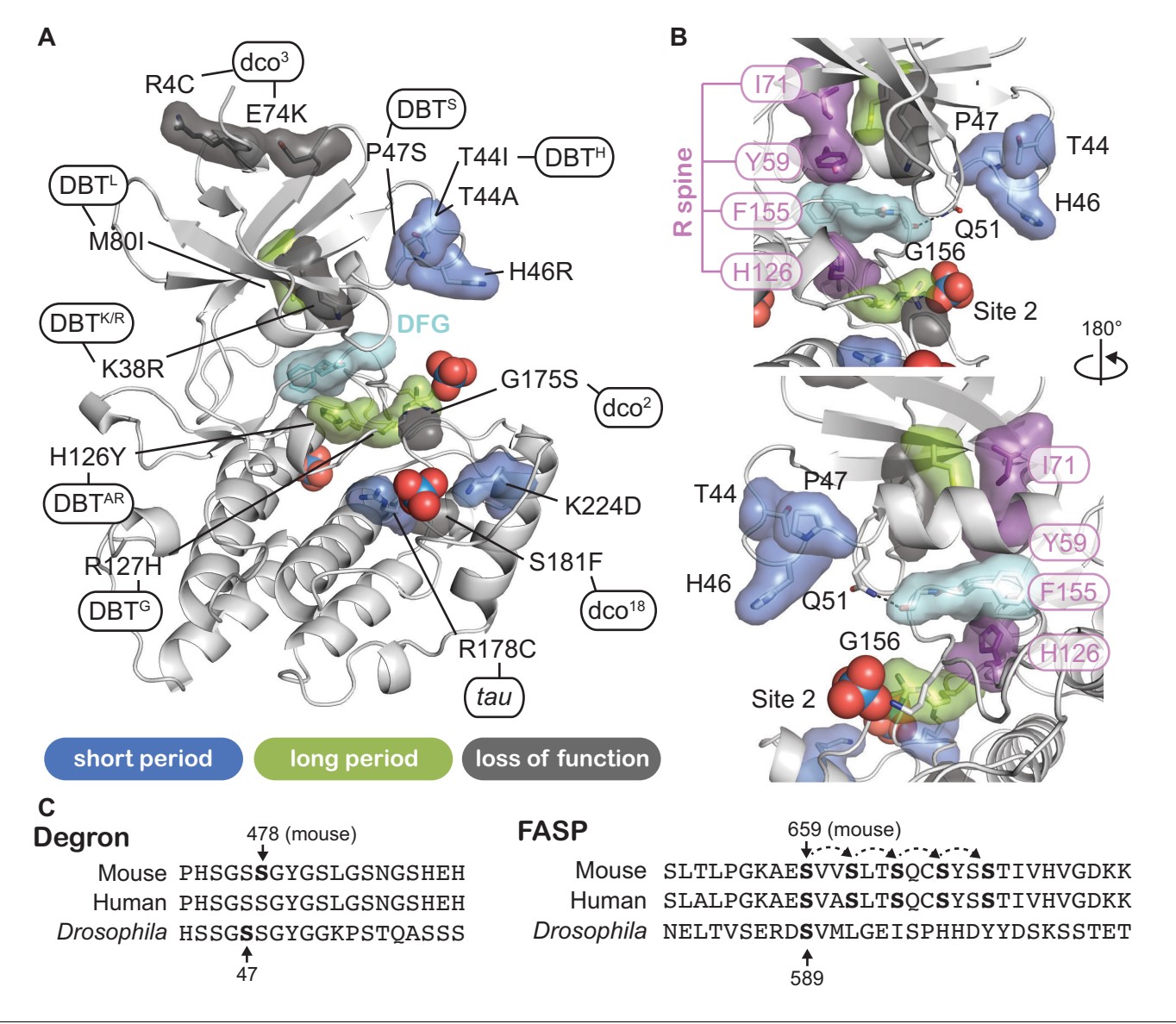

**Figure 6.** Proximity of CK1 alleles map to catalytic and substrate binding sites. (**A**) Structure of CK1δ (PDB: 1CKJ, chain B) clock relevant alleles mapped from mammalian CK1 or *Drosophila* DBT. Mutants are colored by phenotype: short period (blue), long period (green), loss of function (gray). DFG catalytic motif, cyan. (**B**) View of N-lobe period mutants and the regulatory spine (R-spine, purple) with F155 of the DFG motif in cyan. Polar interactions between Q51 and G156 that link the N- and C-lobe are depicted with a dashed black line. (**C**) Alignment of the mammalian and *Drosophila* Degron and FASP/stabilizing sequences. Residues in bold have experimental support for phosphorylation. Dashed arrows indicate sequential phosphorylation following the consensus pSxxS motif.

The online version of this article includes the following figure supplement(s) for figure 6:

**Figure supplement 1.** Conservation of CK1δ/ε in eukaryotes.

(*Figure 7B*), although not to the same degree as *tau*. On balance, it appears that mutants near Site two might act differently from *tau* in that they retain FASP priming activity while increasing activity at the Degron relative to WT.

We also studied another short period mutant located at Site 1, K224D, which was recently reported to have a ~ 20 hr circadian period like *tau* (*Shinohara et al., 2017*). Unlike *tau*, K224D retained FASP priming activity in HEK293 cells (*Figure 7C*), although it exhibited a sharp increase in Degron phosphorylation similar to *tau* (*Figure 7D*). This effect was not dependent on the charge

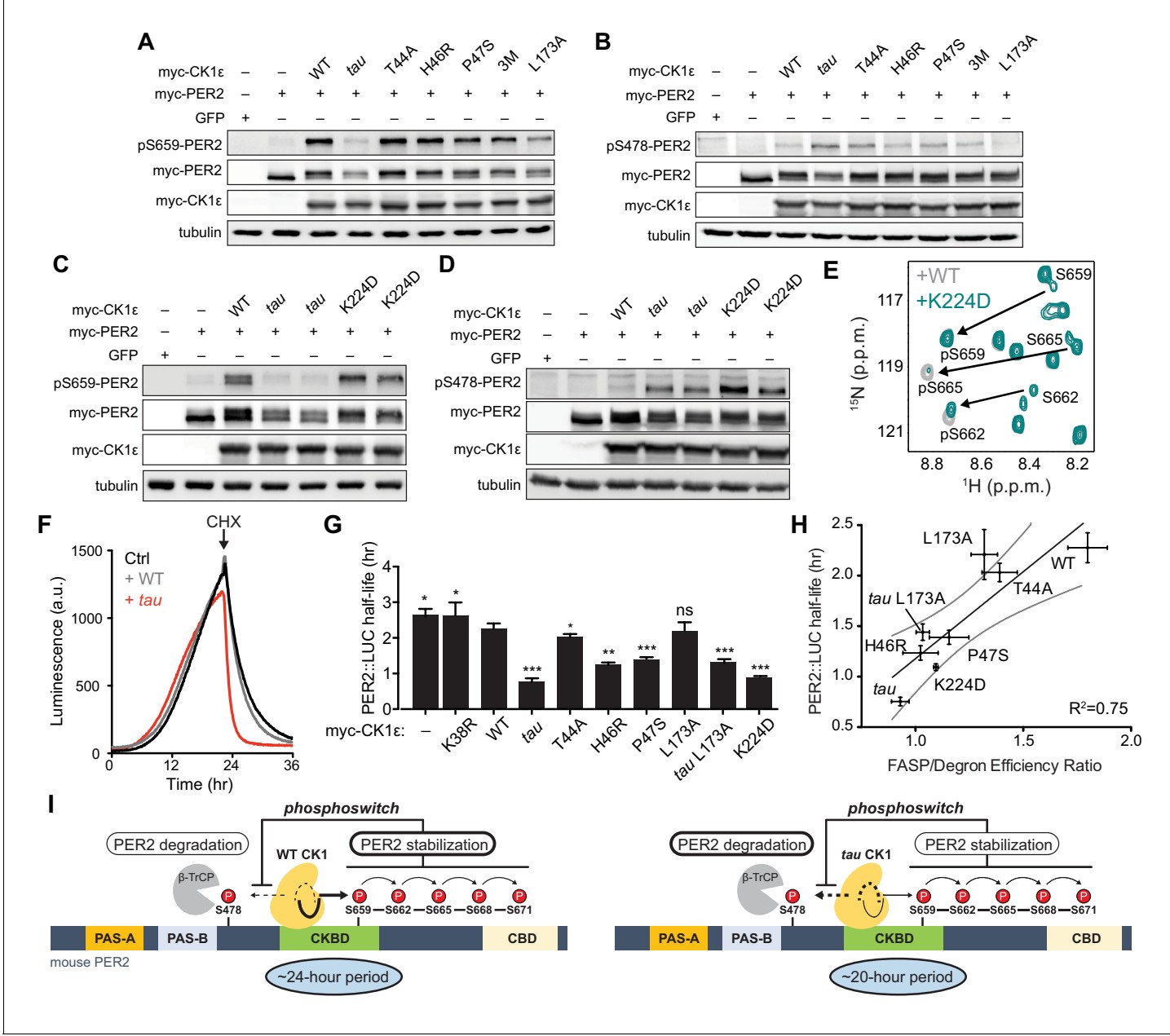

**Figure 7.** Substrate discrimination on the PER2 phosphoswitch is regulated by the CK1 activation loop switch. (A,B) Western blot of FASP priming site, detecting pS659 (A) or Degron site, detecting pS478 (B) phosphorylation on mouse myc-PER2 in HEK293 cell lysates after transfection with indicated myc-CK1ε expression plasmids. 3M triple mutant: T44A, H46R, P47S. Representative blot from n = 3 shown. (C,D) Western blot of FASP priming (C) or Degron (D) phosphorylation on PER2 as above with the myc-CK1ε K224D mutant. Representative blot from n = 3 shown with replicate samples loaded for *tau* and K224D. (E) Overlaid $^{15}$N/$^{1}$H HSQC spectra at 3 hr timepoint in the NMR kinase assay on 200 μM $^{15}$N FASP + 1 μM K224D (teal) or WT (gray) CK1δ ΔC. Arrows, phospho-specific peaks corresponding to pS659, pS662, and pS665. (F) Representative real-time luminescence data for PER2::LUC stability in HEK293 cells transfected with myc-PER2::LUC plus empty vector (black) or myc-CK1ε WT (gray) or *tau* (red) as indicated (n = 4). 40 μg/mL cycloheximide (CHX) added 24 hr post-transfection (arrow). (G) Quantification of PER2::LUC half-life with different myc-CK1ε mutants. Data represent mean ± s.d. (n = 4) with significance assessed as above. (H) Scatterplot with linear regression analysis of the ratio of enzyme efficiencies ($k_{cat}$/$K_m$) for FASP and Degron relative to the PER2::LUC half-life determined in panel G and *Figure 7—figure supplement 1*. All data are plotted as mean ± s.d. (n = 4 for PER2::LUC and n = 3–4 for enzyme efficiencies). Black, linear regression to data; gray, 95% confidence interval. (I) The conformational switch of the CK1δ/ε activation loop is coupled to substrate selection in the PER2 phosphoswitch. Left, the activation loop of the WT kinase is stable in the 'loop down' conformation, leading to preferential phosphorylation the FASP region, which stabilizes PER2 by reducing phosphorylation of the Degron. Right, the activation loop of *tau* kinase is better tolerated in the alternate 'loop up' conformation leading to an intrinsic gain of kinase function on the

*Figure 7 continued on next page*

*Figure 7 continued*

Degron and loss of kinase function on the stabilizing FASP region. This switch in substrate preference promotes PER2 degradation and leads to a shorter circadian period. CKBD, CK1 binding domain; CBD, CRY binding domain. See also *Figure 7—figure supplement 1*.

The online version of this article includes the following figure supplement(s) for figure 7:

**Figure supplement 1.** Effect of CK1 mutations on PER2 phosphorylation and stability.

inversion, as we saw the same effect with a K224A mutant (*Figure 7—figure supplement 1*). Despite retaining priming activity in cells, we noticed that the enzyme efficiency of K224D was decreased on the FASP peptide, which contains multiple serines (*Supplementary file 1b*). This suggested that while priming of the FASP site S659 might be intact, subsequent phosphorylation of the downstream sites might be compromised in K224D. To test this, we used the NMR-based kinase assay to provide information on the stepwise phosphorylation of the FASP region (*Narasimamurthy et al., 2018*). Consistent with the cellular data, K224D retained normal priming activity at S659, but phosphorylation of the downstream serines that conform to the pSxxS consensus motif was compromised relative to WT kinase (*Figure 7E*). Therefore, although *tau* and K224D both enhance Degron phosphorylation to a similar degree and lead to ~20 hr circadian periods (*Lowrey et al., 2000*; *Shinohara et al., 2017*), this is likely achieved through different mechanisms on the kinase—*tau* reduces both priming and downstream phosphorylation of the FASP region, while K224D (and K224A) likely just disrupts the phosphorylation of the downstream pSxxS consensus motifs.

## The ratio of FASP/Degron enzyme efficiency correlates with PER2 stability

One commonality of short period mutants is enhanced Degron phosphorylation in cells, as both cellular (*Gallego et al., 2006*; *Vanselow et al., 2006*) and in vivo studies of short period mutants demonstrate that PER2 stability is decreased in a CK1-dependent manner (*Price et al., 1998*; *Xu et al., 2007*). We measured the effect of CK1 mutants on the half-life of PER2::LUC in real-time after cycloheximide treatment (*Figure 7F and* ). All of the mutants, apart from L173A, lead to a shorter half-life of PER2::LUC (*Figure 7G*). We initially mutated L173 because it appears to serve as a latch for the activation loop in both its 'loop up' and 'loop down' conformations (*Figure 2C*). We found that the mutant substantially decreased kinase activity at both the FASP priming and Degron sites in the transfection-based assay and in vitro (*Figure 7A–B* and *Supplementary file 1b*), and a double mutant with *tau* decreased its activity on the Degron to a similar degree (*Figure 7—figure supplement 1*). We hypothesized that the L173A mutant might have a half-life similar to WT because its reduced activity on the stabilizing FASP region was offset to a similar degree on the Degron (*Figure 7G* and *Supplementary file 1b*). Indeed, we observed a significant correlation between PER2::LUC half-life and the intrinsic ratio of CK1 kinase activity on these two key sites that establish the phosphoswitch mechanism (*Figure 7H*). Collectively, these data support a new model for CK1 regulation whereby activation loop dynamics orchestrate substrate specificity and contribute directly to the phosphoswitch regulation of PER2 stability, and therefore, the timing of circadian rhythms (*Figure 7I*).

## Discussion

Despite its powerful control over the timing of circadian rhythms in eukaryotes from humans to green algae (*van Ooijen et al., 2013*; *Xu et al., 2005*), very little is known about the molecular determinants of CK1δ substrate selectivity and activity. We discovered a conformational switch in the activation loop of CK1 that regulates its activity on two regulatory regions in PER2 that control its stability and circadian timing in mammals (*Zhou et al., 2015*). We show that anion binding regulates the conformation of this switch to reshape the substrate binding cleft and thereby control substrate selectivity. Short period mutants in CK1 from *Drosophila* to mammals reveal that all exhibit an intrinsic increase in kinase activity on the Degron, consistent with an earlier observation in cells (*Gallego et al., 2006*). However, the *tau* mutant disrupts the apparent allosteric regulation between anion binding sites; our MD simulations revealed that *tau* alters the dynamics of the activation loop

and adjacent loop L-EF that likely underlie the decreased activity of *tau* on the stabilizing FASP region.

Allosteric regulation is a common feature of protein kinases, often based on dynamic changes in ensembles of residues that can occur in the absence of major changes in conformation (*Kornev and Taylor, 2015*). CK1 has remarkable, histone-like conservation (≥95% identity) of the entire surface-exposed area linking the two key anion binding sites and substrate binding cleft, suggesting that the mechanisms we discovered here likely apply broadly to circadian rhythms as well as other CK1-regulated processes in other eukaryotes. Binding of regulatory anions at these conserved sites could arise from phosphorylated CK1 itself in cis via its autoinhibitory tail (*Rivers et al., 1998*), or in trans from the binding of phosphorylated substrates like the FASP region, to allow for the generation of feedback regulation directly on the kinase. In line with this, changes in the sequence (*Fustin et al., 2018*) or phosphorylation (*Eng et al., 2017*; *Takano et al., 2004*) of the autoinhibitory tail of CK1δ/ε could alter the balance of PER2 phosphorylation at the FASP or Degron to control circadian period. Therefore, our study demonstrates that the dynamics of CK1δ/ε directly encode its activity in the PER2 phosphoswitch (*Zhou et al., 2015*). The conformational equilibrium of the activation loop may also play a role in the temperature-compensated activity of CK1δ/ε observed in vitro that is linked to loop L-EF, an insertion in the clock-relevant kinases CK1δ and CK1ε that plays a role in maintaining its relative insensitivity to temperature (*Shinohara et al., 2017*).

We used PER2 stability here as a cellular proxy to study the effect of mutations in CK1 on circadian timing. PER proteins seem to have a special role in the mammalian clock as state variables that define both the timing and phase of circadian rhythms through changes in their abundance (*Balsalobre et al., 2000*; *Zylka et al., 1998*). CK1-dependent changes in PER abundance likely affect circadian period based on their role as stoichiometrically limiting factors in the assembly of transcriptional repressive complexes in the feedback loop of the molecular clock (*Aryal et al., 2017*; *Kim and Forger, 2012*; *Lee et al., 2011b*). Recent studies have shown that CK1 may allow target other clock proteins for phosphorylation in the repressive complexes (*Aryal et al., 2017*), a property that seems to be conserved in *Drosophila* (*Yu et al., 2006*). Therefore, more studies are needed to fully understand the interplay between PER2 stability, PER-CK1 interactions and its regulation by post-translational modifications, including by phosphatases (*Lee et al., 2011a*) and other kinases (*Hayasaka et al., 2017*; *Hirota and Kay, 2009*; *Oshima et al., 2019*) that ultimately control circadian rhythms.

There is still much more to be learned about the factors that dictate CK1 substrate selectivity. CK1 is clearly highly active on primed (pSxxS) sites (*Narasimamurthy et al., 2018*), supporting its designation as the canonical consensus motif of the kinase. However, we find it compelling that many of the biologically important roles of CK1δ/ε and the related kinase CK1α as key regulators of Wnt signaling (*Marin et al., 2003*), the DNA damage response (*Knippschild et al., 1997*), cell cycle (*Penas et al., 2015*), and circadian rhythms (*Kloss et al., 1998*; *Lowrey et al., 2000*) depend on their activity at non-consensus sites. The ability of these kinases to phosphorylate lower affinity, non-consensus sites on PER2 is likely dependent on the formation of a stable, stoichiometric complex with PER2 via highly conserved sites that flank the FASP phosphorylation region in the CKBD (*Eide et al., 2005*; *Lee et al., 2004*). The generally low activity of CK1 that we observe on clock-relevant non-consensus sequences may also be important for the slow timescale of circadian rhythms. This property is also conserved in KaiC, the enzyme that controls circadian timing in the cyanobacterial system (*Abe et al., 2015*), suggesting commonalities in the biochemical origins of building slow biological clocks.

## Materials and methods

### Cell lines

HEK293 cells were purchased from American Type Culture Collection (Cat.# CRL-1573, RRID:CVCL_0045). Cells were authenticated by STR profiling and routinely tested for mycoplasma using commercial kits.

Cell culture, reagents and transfection myc-mPer2, myc-mPer2 S659A, mPer2::Luc, and myc-CK1ε expression plasmids were described previously (*Eide et al., 2005*; *Eng et al., 2017*; *Zhou et al.,*

*2015*). Mutations of the kinase domain were introduced by Quikchange site-directed mutagenesis (Agilent) and validated by sequencing.

HEK293 were cultured in Dulbecco's Modified Eagle's Medium (DMEM, Gibco) supplemented with 10% FBS (Gibco), 50 units/mL penicillin, 50 µg/mL streptomycin (Invitrogen) and maintained at 37°C in a 5% $CO_2$ environment. Cells were transfected using Lipofectamine 2000 transfection reagent (Life Technologies) following the manufacturer's instructions. For transfections titrating expression of myc-CK1ε, either 10 or 50 ng of plasmid was used; total plasmid DNA of either 1 or 2 µg was used for each well of a 12 or 6-well-plate respectively.10 µM MG132 was added to cultures 24 hr prior to harvest to prevent proteasomal degradation for experiments shown in *Figure 7A and B*.

## PER2::LUC half-life measurement

Mouse PER2::LUC expression plasmids (10 ng) were transiently transfected alone or with myc-CK1ε (100 ng) in 35 mm dishes of HEK293 cells in phenol red-free DMEM in the presence of 100 mM D-luciferin (122799, PerkinElmer), 10 mM HEPES and 1.2 g/L sodium bicarbonate. Dishes were sealed with 40 mm cover glasses and vacuum grease, and incubated in the LumiCycle (Actimetrics). The next day, 40 µg/mL cycloheximide (Sigma) was added per 35 mm dish. Luminescence data were used to calculate PER2::LUC half-life in Prism (GraphPad) using one-phase decay algorithm as described previously (*Zhou et al., 2015*). Briefly, half-lives were calculated using the one-phase decay algorithm in Prism (GraphPad) using the raw luciferase activity, beginning from the point of cycloheximide addition to the plateau at minimum luciferase activity (n = 4).

## SDS-PAGE and western blotting

Whole cell extracts of transfected HEK293 cells lysed on ice with cell lysis buffer (50 mM Tris-HCl pH 8.0, 150 mM NaCl, 1% (vol/vol) Nonidet P-40% and 0.5% deoxycholic acid containing Complete Protease Inhibitors (Roche) and PhosStop Phosphatase Inhibitors (Roche)) were analyzed by denaturing SDS-PAGE gel, which was transferred on PVDF membrane (Immobilon, Millipore). The blot was probed using the indicated primary antibodies: anti-myc (9E10) (sc-40, Santa Cruz Biotechnology; RRID:AB_627268) and anti-tubulin (ab52623, Abcam; RRID: AB_869991) were purchased from commercial providers, while rabbit polyclonal antibodies were generated against phospho-Ser478, phospho-Ser659 or phospho-Ser662 of mouse PER2 and purified against the phosphopeptides by Abfrontier (Young In Frontier Co.). The phosphopeptides for phospho-Ser478 and phospho-Ser659 have been described elsewhere (*Narasimamurthy et al., 2018*; *Zhou et al., 2015*); the phosphopeptide KAESVVpSLTSQ-Cys was used to generate the phospho-Ser662 antibody. HRP-conjugated goat secondary antibodies for anti-rabbit (1706515, Bio-Rad) and anti-mouse (1706516, Bio-Rad) were with standard ECL reagents (Thermo Fisher Scientific). Densitometric analysis of western blot bands was performed using ImageJ software (National Institutes of Health).

## Expression and purification of recombinant proteins

All proteins were expressed from a pET22-based vector in *Escherichia coli* Rosetta2 (DE3) cells based on the Parallel vector series (*Sheffield et al., 1999*). The wild-type recombinant FASP peptide (residues 645–687) or Degron peptide (residues 475–505) were cloned from human PER2, and a short wild-type mouse FASP peptide (residues 642–666) all contain an N-terminal WRKKK polybasic motif for in vitro kinase assays and a tryptophan for UV detection during purification. All peptides were expressed downstream of an N-terminal TEV-cleavable His-NusA tag. Human CK1δ catalytic domains (CK1δ ΔC, residues 1–317) were all expressed in Rosetta2 (DE3) cells (Sigma Aldrich) with a TEV-cleavable His-GST tag. Mutations were made using standard site-directed mutagenesis protocols and validated by sequencing. All proteins and peptides expressed from Parallel vectors have an additional N-terminal vector artifact of 'GAMDPEF' remaining after TEV cleavage. Cells were grown in LB media (for natural abundance growths) or M9 minimal medium with the appropriate stable isotopes ($^{15}N$, $^{13}C$ for NMR, as done before *Narasimamurthy et al., 2018*) at 37°C until the O.D.$_{600}$ reached ~0.8; expression was induced with 0.5 mM IPTG, and cultures were grown for approximately 16–20 hr more at 18°C.

For CK1δ kinase domain protein preps, cells were lysed in 50 mM Tris pH 7.5, 300 mM NaCl, 1 mM TCEP, and 5% glycerol using a high-pressure extruder (Avestin). HisGST-CK1δ ΔC fusion

proteins were purified using Glutathione Sepharose 4B resin (GE Healthcare) using standard approaches and eluted from the resin using Phosphate Buffered Saline with 25 mM reduced glutathione. His$_6$-TEV protease was added to cleave the His-GST tag from CK1δ ΔC at 4°C overnight. Cleaved CK1δ ΔC was further purified away from His-GST and His-TEV using Ni-NTA resin (Qiagen) and subsequent size exclusion chromatography on a HiLoad 16/600 Superdex 75 prep grade column (GE Healthcare) in 50 mM Tris pH 7.5, 200 mM NaCl, 5 mM BME, 1 mM EDTA, and 0.05% (vol/vol) Tween 20. Purified CK1δ ΔC proteins used for in vitro kinase assays were buffer exchanged into storage buffer (50 mM Tris pH 7.5, 100 mM NaCl, 1 mM TCEP, 1 mM EDTA, and 10% glycerol) using an Amicron Ultra centrifugal filter (Millipore) and frozen as small aliquots in liquid nitrogen for storage at -80°C.

For PER2 peptide preps, cells were lysed in 50 mM Tris pH 7.5, 500 mM NaCl, 2 mM TCEP, 5% glycerol and 25 mM imidazole using a high-pressure extruder (Avestin). His-NusA-FASP or His-NusA-Degron fusion proteins were purified using Ni-NTA resin using standard approaches and eluted from the resin using 50 mM Tris pH 7.5, 500 mM NaCl, 2 mM TCEP, 5% glycerol and 250 mM imidazole. His-TEV protease was added to cleave the His$_6$-NusA tag from the PER2 peptides at 4°C overnight. The cleavage reaction was subsequently concentrated and desalted into low imidazole lysis buffer using a HiPrep 26/10 Desalting column. Peptides were purified away from His-NusA and His-TEV using Ni-NTA resin with 50 mM Tris pH 7.5, 500 mM NaCl, 2 mM TCEP, 5% glycerol and 25 mM imidazole. Peptides were purified by size exclusion chromatography on a HiLoad 16/600 Superdex 75 prep grade column, using NMR buffer (25 mM MES pH 6.0, 50 mM NaCl, 2 mM TCEP, 1 mM EDTA, 11 mM MgCl$_2$) or 1x kinase buffer (25 mM Tris pH 7.5, 100 mM NaCl, 10 mM MgCl$_2$, and 2 mM TCEP) for NMR or ADP-Glo kinase assays, respectively.

## Radioactive and ELISA-based kinase assays

Mouse PER2 peptides (primed FASP, RKKKTEVSAHLSSLTLPGKAEpSVVSLTSQ, or unprimed FASP, RKKKTEVSAHLSSLTLPGKAESVVSLTSQ, or the Degron, RKKKPHSGSSGYGSLGSNGSHEHMSQTSSSD SN, from *Isojima et al., 2009*) and the CK1tide peptide (KRRRALpSVASLPGL, from *Isojima et al., 2009*; *Shinohara et al., 2017*) were synthesized and purified to 95% or higher (SABio).

For the radioactive kinase assay, two independent reaction mixtures of 50 µL containing 200 µM of the FASP or Degron peptides in reaction buffer (25 mM Tris pH 7.5, 7.5 mM MgCl$_2$, 1 mM DTT, 0.1 mg/mL BSA) were preincubated for 5 min with or without 20 nM CK1δ ΔC (for primed FASP substrate) or 200 nM CK1δ ΔC (for unprimed FASP or Degron) and the reaction was started by addition of 750 µM of UltraPure ATP (Promega) containing 1–2 µCi of γ-$^{32}$p ATP (Perkin Elmer). After incubation of the reaction mix at 30°C, an 8 µL aliquot of the reaction mix was transferred to P81 phosphocellulose paper (Reaction Biology Corp) at the indicated timepoints. The P81 paper was washed three times with 75 mM of orthophosphoric acid and once with acetone. The air-dried P81 paper was counted for P$_i$ incorporation using a scintillation counter (Perkin Elmer) by Cherenkov counting. Results shown are from four independent assays.

For the ELISA kinase assay, unprimed FASP peptide was diluted to 2 µg/mL in Carbonate buffer, pH 9.5 (0.1 M sodium carbonate) and coated onto a 96 well plate (100 µL/well). The next day, wells were washed three times with wash buffer (PBS with 0.05% Tween, PBS-T) and once with kinase buffer (25 mM Tris pH 7.5, 5 mM beta glycerol phosphate, 2 mM DTT and 0.1 mM sodium orthovanadate). Reaction mixture (50 µL) containing 10 ng of CK1δ ΔC purified protein in the kinase buffer including 10 mM MgCl$_2$ and 200 µM ATP was added onto each well and the plate was incubated at 30°C for 1 hr. Next, the reaction mixture was removed and the wells were washed with three times with wash buffer and incubated with blocking buffer (PBS-T with 5% BSA) for 1 hr at room temperature. Subsequently, wells were incubated with pS659 Ab, anti-rabbit antibody conjugated to Biotin and Streptavidin-HRP for 1 hr at room temperature with a washing step after each incubation as above. For signal detection, TMB (1-Step Ultra TMB-ELISA, Thermo Scientific) was added, incubated for color development at room temperature and stopped with the addition of STOP solution (Thermo Scientific). The plate was read at 450 nm using an xMark Spectrophotometer plate reader (Biorad). Results shown are from four independent assays.

## NMR-based kinase assay

NMR spectra were collected on a Varian INOVA 600 MHz or a Bruker 800 MHz spectrometer equipped with a $^1$H, $^{13}$C, $^{15}$N triple resonance z-axis pulsed-field-gradient cryoprobe. Spectra were processed using NMRPipe (*Delaglio et al., 1995*) and analyzed using CCPNmr Analysis (*Vranken et al., 2005*). Backbone assignments were obtained previously for the mouse FASP (BMRB entry: 27306) (*Narasimamurthy et al., 2018*). NMR kinase reactions were performed at 30°C with 0.2 mM $^{15}$N-mouse FASP and/or Degron, 2.5 mM ATP and 1 µM CK1d ΔC (WT or *tau*). SOFAST HMQC spectra (total data acquisition = 6 min) were collected at the indicated intervals for 3 hr and relative peak volumes were calculated and normalized as described previously (*Narasimamurthy et al., 2018*). Data analysis was performed using Prism (GraphPad), with data fit to either a one-phase exponential or linear regression.

## ADP-Glo kinase assay

Kinase reactions were performed on the indicated recombinant peptides (Degron or FASP) using the ADP-Glo kinase assay kit (Promega) according to manufacturer's instructions. All reactions were performed in 30 µL volumes using 1x kinase buffer (25 mM Tris pH 7.5, 100 mM NaCl, 10 mM MgCl$_2$, and 2 mM TCEP) supplemented with ATP and PER2 substrate peptides as indicated. To determine the apparent $2^{nd}$-order rate constants, triplicate reactions containing 10 µM substrate, 100 µM ATP, and 0.2 µM CK1δ ΔC kinase were incubated in 1x kinase buffer at room temperature for 3 hr (and repeated for n = 3 independent assays). Linearity of the reaction rate with respect to time was determined by performing larger reactions (50 µL) with wild-type and *tau* CK1δ ΔC and either the FASP or Degron substrate; 5 µL aliquots were taken and quenched with ADP-Glo reagent at discrete time points up to 3 hr (data not shown). Luminescence measurements were taken at room temperature with a SYNERGY2 microplate reader in 384-well microplates. Data analysis was performed using Excel (Microsoft) or Prism (GraphPad).

## Statistical analyses

All statistical analyses were done using Prism (GraphPad). p-values were calculated using unpaired two-tailed Students t-tests. In all figures, * indicates $p < 0.05$, **$p < 0.01$, ***$p < 0.001$, ****$p < 0.0001$; ns, not significant.

## Crystallization and structure determination

Crystallization was performed by hanging-drop vapor-diffusion method at 22°C by mixing an equal volume of CK1δ ΔC with reservoir solution. The reservoir solution for CK1δ ΔC *tau* (R178C) (8.5 mg/mL) was 50 mM sodium acetate pH 6.0, 300 mM ammonium sulfate, and 17.5% (vol/vol) PEG 2000. The reservoir solution for crystallizing CK1δ ΔC wild-type (8 mg/mL) without sulfate was 150 mM succinic acid pH 5.5% and 17% (vol/vol) PEG 3350. The reservoir solution for CK1δ ΔC K171E (7.7 mg/mL) was 50 mM sodium acetate pH 6.0, 350 mM ammonium sulfate, and 17.5% (vol/vol) PEG 3500. The crystals were looped and briefly soaked in a drop of reservoir solution and then flash-cooled in liquid nitrogen for X-ray diffraction data collection. For CK1δ ΔC *tau*, a cryopreservant of reservoir solution with 20% (vol/vol) glycerol was used. Data sets were collected at the APS beamline 23-ID-D, and the ALS beamline 8.3.1. Data were indexed, integrated and merged using the CCP4 software suite (*Winn et al., 2011*). Structures were determined by molecular replacement with Phaser MR (*McCoy et al., 2007*) using the ADP-bound structure of wild-type CK1δ ΔC (PDB: 5 × 17). Model building was performed with Coot (*Emsley et al., 2010*) and structure refinement was performed with PHENIX (*Adams et al., 2011*). All structural models and alignments were generated using PyMOL Molecular Graphics System 2.0 (Schrödinger).

## Molecular dynamics

### Initial structures

To simulate *tau* CK1δ, we used the crystallographic structure reported herein (PDB: 6PXN), using chain A to simulate the 'loop up' conformation and chain B to simulate the 'loop down' conformation. As starting structures for the WT CK1δ simulations, we selected an apo structure (PDB: 1CKJ) of 2.46 Å resolution, which has the activation loop crystallized both in 'up' (chain A) and 'down' (chain B) conformations (*Longenecker et al., 1996*). This structure contained two (chain A) or three

(chain B) $WO_4^{-2}$ anions, which were computationally replaced by $SO_4^{2-}$ anions at the same positions, to make the simulations of WT CK1δ more comparable to the *tau* simulations. The initial backbone conformation of loop L-EF was similar in all of the structures used as starting points for the MD simulations with a backbone RMSD for residues 210 to 230 of <3.7 Å between initial structures. Using these structures, we created the systems to be simulated, described in *Supplementary file 1d*.

## Systems set up and equilibration

The systems described in *Supplementary file 1d* were refined by (i) position restrained energy minimization followed by (ii) full energy minimization, using Maestro (Schrödinger). The protonation states of the minimized models were estimated using the H++ server (*Anandakrishnan et al., 2012*) and hydrogens were added using pdb2pqr (*Dolinsky et al., 2007*).

All systems were solvated in a pre-equilibrated cubic TIP3P (*Jorgensen et al., 1983*) water box with at least 15 Å between the protein and the box boundaries. The net charge of the system was neutralized with $Na^+$ or $Cl^-$ counterions. Parameters for protein atoms and counterions were extracted from the ff14SB forcefield (*Maier et al., 2015*), while parameters for the $SO_4^{2-}$ anions were extracted from the Generalized Amber Force Field (GAFF) (*Wang et al., 2004*) and adjusted as proposed by *Kashefolgheta and Vila Verde (2017)*.

Minimization and equilibration were performed with AMBER 16 (*Case et al., 2016*), using the following protocol: (i) 2000 steps of energy minimization with a 500 kcal $mol^{-1}$ $Å^{-1}$ position restraint on protein and $SO_4^{2-}$ anions; (ii) 1000 steps of energy minimization with a 500 kcal $mol^{-1}$ $Å^{-1}$ position restraint on protein atoms only; (iii) 2000 steps of energy minimization without position restraints; (iv) 50 ps of NVT simulation, with gradual heating to a final temperature of 300 K, with 10 kcal $mol^{-1}$ $Å^{-1}$ position restraint on protein and $SO_4^{2-}$ anions; (v) 1 ns of NPT simulation to equilibrate the density (or final volume of the simulation box).

## Conventional (cMD) and Gaussian accelerated MD (GaMD) simulations

Before running the GaMD simulations, we ran 100 ns of conventional MD simulations for each system, using AMBER 16 (*Case et al., 2016*) These simulations were performed in the NVT regime, with a time step of 2 fs, and all bonds involving hydrogen atoms were restrained with SHAKE (*Ryckaert et al., 1977*). The PME method (*Darden et al., 1993*) was used to calculate electrostatic interaction using periodic boundary conditions, and a 12 Å cutoff was used to truncate non-bonded short-range interactions.

The final conformations produced by the cMD simulations were used as starting configurations for the GaMD simulations, which were performed with AMBER 17 (*Case et al., 2017*). For these simulations, additional acceleration parameters were used to boost the exploration of the conformational space, as described in *Miao et al. (2015)*. All systems had a threshold energy E = $V_{max}$ and were subjected to a dual boost acceleration of both the dihedral and the total potential energies. To optimize the acceleration parameters we first ran 2 ns of MD simulations with no boost potential, during which the minimum, maximum, average and standard deviation ($V_{min}$, $V_{max}$, $V_{av}$, $\sigma_{avg}$) of the total potential and dihedral energies were estimated and used to derive boost potentials as detailed in *Miao et al. (2015)*. These potentials were used to start 50 ns of Gaussian accelerated MD simulations, during which the boost statistics and boost potentials were updated until the maximum acceleration was achieved. The maximum acceleration was constrained setting the upper limit of the standard deviation of the total boost potential to be 6 kcal/mol.

We ran 5 replicas of production GaMD simulations for each system with fixed acceleration parameters derived from the previous equilibration stage. Each replica started from the same initial conformation, but the atoms were given different initial velocities, consistent with a Maxwell-Boltzmann distribution at 300 K. Each production simulation ran for 500 ns, totalizing 2.5 µs of sampling for each system, and 15 µs in total.

## Analysis of GaMD simulations

### Conformational dynamics of loops

The RMSD of the activation loop (*Figure 4A–D*) or loop L-EF (*Figure 4E–H*) was measured throughout the GaMD trajectories with CPPTRAJ using the *rms* module (*Roe and Cheatham, 2013*).

## Volumetric analysis of the substrate binding cleft

The volume and shape of the substrate binding cleft and adjacent anion binding sites were obtained from the GaMD trajectories using POVME 3.0 (*Wagner et al., 2017*). For each trajectory, conformations of WT or *tau* CK1δ were extracted every two ns, stripping all water molecules, counter-ions and sulfate anions. All protein conformations were superimposed to the same reference frame and the substrate binding cleft and adjacent anion binding sites were encompassed by three overlapping spheres as shown in *Figure 4—figure supplement 1*. POVME was then used to estimate the free (or empty) volume within the three overlapping spheres for all conformations extracted from the GaMD trajectories. The resulting volumes were averaged to produce three-dimensional density maps, as shown in *Figure 4* (panels I-L). The maps in *Figure 4* were contoured at 0.10 and represent regions that are found more frequently 'open' during the simulations.

## Principal modes of motion

To detect the principal modes of motion displayed by WT and *tau* CK1δ, we used atomic fluctuations sampled during the GaMD simulations to perform Principal Component Analysis (PCA) (*Amadei et al., 1993*; *Amadei et al., 1996*). Before performing PCA, we stripped the trajectories of all solvent, ions and protein side-chains, keeping the backbone atoms only. We then concatenated and aligned these new trajectories to the same reference frame. To construct and diagonalize the co-variance matrix of atomic fluctuations, we used the *matrix* and *analyze* modules, respectively, in CPPTRAJ (*Roe and Cheatham, 2013*). For each system (WT CK1δ$^{loop\ down}$, WT CK1δ$^{loop\ up}$, *tau* CK1δ$^{loop\ down}$, *tau* CK1δ$^{loop\ up}$), we projected their respective trajectories into the obtained eigenvector space using the *projection* function in CPPTRAJ (*Roe and Cheatham, 2013*). The projections of each system along the subset of the eigenvector space formed by the 1$^{st}$ and 2$^{nd}$ principal components are shown in *Figure 5* and *Figure 5—figure supplement 1*.

All visualization of the GaMD simulations was performed with VMD (*Humphrey et al., 1996*).

## Acknowledgements

We would like to thank Danny Forger, Jae Kyoung Kim, Yinglong Miao, and J Andrew McCammon for useful discussions and Sivakumar Parthiban for technical assistance. We also thank the beamline staff for their assistance at the Advanced Photon Source beamline 23-ID-D and Advanced Light Source beamline 8.3.1, as well as the San Diego Supercomputer Center (SDSC) for technical support. This work was funded by the National Medical Research Council of Singapore Grant NMRC/CIRG/1465/2017 (to DMV) and National Institutes of Health Grants R01 GM031749 (to CGR), GM107069 and R01 GM121507 (to CLP), as well as funds from the NIH Office of the Director under Award S10 OD018455 for the 800 MHz NMR spectrometer used here. S Hunt was supported by NRSA F32 GM133149.

## Additional information

### Funding

| Funder | Grant reference number | Author |
|---|---|---|
| National Institutes of Health | GM107069 | Carrie L Partch |
| National Institutes of Health | GM121507 | Carrie L Partch |
| National Institutes of Health | OD018455 | Carrie L Partch |
| National Medical Research Council | NMRC/CIRG/1465/2017 | David M Virshup |
| National Institutes of Health | GM031749 | Clarisse G Ricci |
| National Institutes of Health | GM133149 | Sabrina R Hunt |

The funders had no role in study design, data collection and interpretation, or the decision to submit the work for publication.

## Author contributions
Jonathan M Philpott, Conceptualization, Formal analysis, Validation, Investigation, Visualization; Rajesh Narasimamurthy, Conceptualization, Formal analysis, Validation, Investigation; Clarisse G Ricci, Conceptualization, Formal analysis, Investigation, Visualization; Alfred M Freeberg, Sabrina R Hunt, Lauren E Yee, Rebecca S Pelofsky, Investigation; Sarvind Tripathi, Validation; David M Virshup, Carrie L Partch, Conceptualization, Supervision, Funding acquisition, Visualization, Project administration

## Author ORCIDs
Jonathan M Philpott (iD) https://orcid.org/0000-0002-0793-0193
Rajesh Narasimamurthy (iD) https://orcid.org/0000-0003-4224-3791
Clarisse G Ricci (iD) https://orcid.org/0000-0002-3289-2248
Alfred M Freeberg (iD) https://orcid.org/0000-0003-0365-5769
Lauren E Yee (iD) https://orcid.org/0000-0002-4278-6115
Rebecca S Pelofsky (iD) http://orcid.org/0000-0001-7944-3880
Sarvind Tripathi (iD) http://orcid.org/0000-0002-6959-0577
David M Virshup (iD) https://orcid.org/0000-0001-6976-850X
Carrie L Partch (iD) https://orcid.org/0000-0002-4677-2861

## Decision letter and Author response
Decision letter https://doi.org/10.7554/eLife.52343.sa1
Author response https://doi.org/10.7554/eLife.52343.sa2

# Additional files

## Supplementary files
• Supplementary file 1. Details of CK1 crystallography, enzyme kinetics, and simulated systems. (**A**) X-ray crystallography data collection and refinement statistics. (**B**) Enzymatic efficiency of CK1δ ΔC (wild-type and mutants). (**C**) Survey of anion binding and activation loop conformation in CK1 family member structures. (**D**) Details of simulated systems.

• Supplementary file 2. CK1 family alleles and their circadian phenotypes.

• Transparent reporting form

## Data availability
Diffraction data have been deposited in the PDB under the accession codes 6PXN, 6PXO, 6PXP.

The following datasets were generated:

| Author(s) | Year | Dataset title | Dataset URL | Database and Identifier |
|---|---|---|---|---|
| Jonathan M Philpott, Rajesh Narasimamurthy, Clarisse G Ricci, Alfred M Freeberg, Sabrina R Hunt, Lauren E Yee, Rebecca S Pelofsky, Sarvind Tripathi, David M Virshup, Carrie L Partch | 2020 | Human Casein Kinase 1 delta (anion-free crystallization conditions) | https://www.rcsb.org/structure/6PXO | RCSB Protein Data Bank, 6PXO |
| Jonathan M Philpott, Rajesh Narasimamurthy, Clarisse G Ricci, Alfred M Freeberg, Sabrina R Hunt, Lauren E Yee, Rebecca S Pelofsky, Sarvind Tripathi, David M Virshup, Carrie L Partch | 2020 | Human Casein Kinase 1 delta Tau mutant (R178C) | https://www.rcsb.org/structure/6PXN | RCSB Protein Data Bank, 6PXN |

Jonathan M Philpott, Rajesh Narasimamurthy, Clarisse G Ricci, Alfred M Freeberg, Sabrina R Hunt, Lauren E Yee, Rebecca S Pelofsky, Sarvind Tripathi, David M Virshup, Carrie L Partch

| | | | | |
|---|---|---|---|---|
| | 2020 | Human Casein Kinase 1 delta Site 2 mutant (K171E) | https://www.rcsb.org/structure/6PXP | RCSB Protein Data Bank, 6PXP |

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
