## [Decision Letter]

**Acceptance summary:**

PERIOD or PER proteins are central components of our biological clock, with a transcription and degradation rhythm that normally oscillates in a cycle of 24 hours. Casein Kinase 1 (CK1) phosphorylates both the Degron and the FASP regions of PER-the former initiates PER degradation and the latter antagonistically stabilizes PER. This study for the first time characterized CK1 activity on its biological substrate in vitro. The results showed that the *tau* mutation of CK1, which shortens the oscillation cycle to about 20 hours, increases the Degron activity of CK1 while decreasing the FASP activity. Further, they showed that disruption of CK1 anion binding, altered conformational dynamics, and a reshaped substrate binding site of CK1 underlie the altered substrate selectivity of CK1, thus providing a molecular mechanism for the circadian disorder associated with *tau* mutation.

**Decision letter after peer review:**

Thank you for submitting your article "Casein kinase 1 dynamics underlie substrate selectivity and the PER2 circadian phosphoswitch" for consideration by *eLife*. Your article has been reviewed by three peer reviewers, one of whom is a member of our Board of Reviewing Editors, and the evaluation has been overseen by Jonathan Cooper as the Senior Editor. The reviewers have opted to remain anonymous.

The reviewers have discussed the reviews with one another and the Reviewing Editor has drafted this decision to help you prepare a revised submission.

Summary:

Casein kinase 1 (CK1) is an important kinase that regulates circadian cycles. It stabilizes Period 2 (PER2) protein by phosphorylating the FASP region but also promotes PER2 degradation by phosphorylating the Degron region. Based on biochemistry analyses, this manuscript reports the effect of *tau* mutation of CK1 on its selectivity: it reduces CK1 activity on the FASP region and enhances CK1 activity on the Degron. Crystal structures and molecular dynamics simulations suggests a CK1 allosteric network connecting the Site 1 and Site 2 anion binding sites with the activation loop. The altered phosphorylation site selectivity of CK1 was explained by *tau* mutation disrupting anion binding of CK1 and, shifting the balance between two possible conformations of the activation loop. The reviewers agree that this is overall a thoughtfully designed and well-executed study.

Essential revisions:

1) It is clear from the kinase assays that CK1 (WT or *tau*) activity on PER2 depends on phosphorylation of PER2 priming site. Have the authors modeled the changes in terms of the secondary structure of the peptide upon phosphorylation at the priming site Ser569? The chemical shift of the peptide upon Ser569 phosphorylation can be used to inform the modeling and to explain why the priming is required.

2) It remains unexplained how the altered activation loop conformational dynamics in *tau* favors degron phosphorylation. Does this study suggest that there are substrate-specific active conformations of CK1? The reviewers suggest a docking study of the degron peptide on CK1 to help understand the underlying mechanism.

3) In the presence of both FASP and degron peptide in vitro, does *tau* still phosphorylate the degron sequence more than WT? What degron residues are crucial to the *tau* propensity for the degron. A kinase assay with peptides that are variants of the degron sequence may help explain why *tau* prefers the degron over FASP. Has the affinity of the degron and FASP peptides been measured for CK1?

4) Perhaps a missed opportunity for the authors is the discussion of the Y225 during structural and MD simulation analysis. This side chain seem to adapt different poses in the loop up and loop down conformations and seem to be a direct structural bridge forming a network consisting (R178-K224) -L173. The reviewers suggest the authors to analyze the motion of Y225 and its coupling to the motion of the activation loop in WT and *tau* simulations. If a strong coupling signal was found, perhaps mutational analysis of this residue could help establish the full molecular allosteric network.

5) This manuscript has room for improvement in terms of readability. The article is wordy and a lot of literature information and previous discussions are presented throughout the whole manuscript due to perhaps authors strong sense of due diligence. By trying to move as much indirect material as possible to the supplementary and balancing the discussion of the previous literature to the more through atomistic detail analysis of the allosteric network, the manuscript can become more readable. Some key terms are not explicitly introduced in the Introduction. For instance, the background of the *tau* mutation is not described in the Introduction, although it is later in the Results.

---

## [Author Response]

Essential revisions:1) It is clear from the kinase assays that CK1 (WT or tau) activity on PER2 depends on phosphorylation of PER2 priming site. Have the authors modeled the changes in terms of the secondary structure of the peptide upon phosphorylation at the priming site Ser569? The chemical shift of the peptide upon Ser569 phosphorylation can be used to inform the modeling and to explain why the priming is required.

To address this concretely, we measured chemical shifts for the Cα and Cβ atoms from a ^13^C/^15^N/^1^H HNCACB spectrum of the phosphorylated FASP and compared these to the Chemical Shift Index (CSI), which reports quantitatively on the propensity for secondary structure (Noor et al., 2015, NAR). Our CSI analysis suggests a modest increase in the propensity for β-strand structure localized within the phosphorylated FASP region (S659-S665 in red, Author response image 1). However, there is recent literature showing that the CSI is biased towards the prediction of secondary structure propensity within phosphorylated regions due to phospho-dependent chemical shift changes (Hendus-Altenburger et al., 2019, J Biomol NMR). In particular, we noted that changes in secondary structure propensity map exclusively to the phosphorylated serine residues within the peptide, with only marginal changes in neighboring residues. Based on these factors and the limited dispersion of proton chemical shifts in the ^15^N/^1^H HSQC spectra, it is highly unlikely that the FASP peptide studied here (33 residues) takes on secondary structure in the presence or absence of phosphorylation.

**Author response image 1. respfig1:** Secondary structure propensity of the FASP peptide.

The larger question of why priming is required for high efficiency activity by CK1 remains to be addressed in future work. We believe that our analysis of the K224D mutant, which likely disrupts anion binding at Site 1 without altering the allosteric regulation at the activation loop like *tau*, suggests that primed peptides likely dock into the anion binding pocket at Site 1 to position the downstream sites. We are actively pursuing modeling and experimental structure determination to probe this model in more detail.

2) It remains unexplained how the altered activation loop conformational dynamics in tau favors degron phosphorylation. Does this study suggest that there are substrate-specific active conformations of CK1? The reviewers suggest a docking study of the degron peptide on CK1 to help understand the underlying mechanism.

We agree with reviewers that the exact mechanism by which altered dynamics of the activation loop favor degron phosphorylation by CK1 remains to be determined. We do believe that our study suggests that there are substrate-specific conformations of CK1. Obviously, obtaining a mechanistic understanding of this will require more in-depth structural studies and/or MD simulations. At the reviewer’s suggestion, we began a series of docking studies of the degron with the wild-type kinase in its loop up or down conformation and have some provocative initial results. We thank the reviewers for their suggestion, but we feel that significantly more work will be needed to validate these initial findings, putting it outside the scope of this study.

*3) In the presence of both FASP and degron peptide* in vitro*, does tau still phosphorylate the degron sequence more than WT? What degron residues are crucial to the tau propensity for the degron. A kinase assay with peptides that are variants of the degron sequence may help explain why tau prefers the degron over FASP. Has the affinity of the degron and FASP peptides been measured for CK1?*

We also think this is an interesting question! Thanks to our NMR-based kinase assay, we can unambiguously resolve phosphorylation on both the FASP and degron substrates in one assay. We ran this experiment in response to the reviewer’s question and found that WT and *tau* CK1 maintain their distinct preferential activity even in the presence of both substrates. These data are described subsection “*tau* exhibits a gain of function on the Degron” and in Figure 1—figure supplement 1G-H.

With regards to the question about the exact specificity determinants on the degron, this is an area of ongoing study in our labs. We are currently working to identify the molecular determinants of CK1 activity on the non-consensus FASP priming site and the degron. We feel that this work is best suited for a more comprehensive report in the future. The affinity of both peptides for the kinase has not been explicitly measured, but we have estimates of the *K*_m_s in the ~10-500 µM range, similar to other kinases.

4) Perhaps a missed opportunity for the authors is the discussion of the Y225 during structural and MD simulation analysis. This side chain seem to adapt different poses in the loop up and loop down conformations and seem to be a direct structural bridge forming a network consisting (R178-K224) -L173. The reviewers suggest the authors to analyze the motion of Y225 and its coupling to the motion of the activation loop in WT and tau simulations. If a strong coupling signal was found, perhaps mutational analysis of this residue could help establish the full molecular allosteric network.

We completely agree with the reviewers and we thank you for pointing out this interesting observation. In going back through our MD simulations, we found that the conformation and mobility of Y225 was indeed different in the tau mutant compared to the WT enzyme. We have included some text in subsection “*tau* stabilizes the rare ‘loop up’ conformation of the CK1 activation loop” to describe this observation and a figure panel in Figure 4—figure supplement 1. In our analysis, we also found evidence that the conformation of Y225 was coupled to the Site 1 anion binding pocket and is therefore potentially in a position to control the equilibrium between the ‘up’ and ‘down’ conformations of the activation loop. However, we feel that a full dissection of the role of Y225 with mutational analysis and functional characterization, as well as a more comprehensive description of the allosteric communication in CK1δ, is beyond the scope of the current study.

5) This manuscript has room for improvement in terms of readability. The article is wordy and a lot of literature information and previous discussions are presented throughout the whole manuscript due to perhaps authors strong sense of due diligence. By trying to move as much indirect material as possible to the supplementary and balancing the discussion of the previous literature to the more through atomistic detail analysis of the allosteric network, the manuscript can become more readable. Some key terms are not explicitly introduced in the Introduction. For instance, the background of the tau mutation is not described in the Introduction, although it is later in the Results.

We agree that the manuscript is wordy. It was a bit challenging presenting the first mechanistic study on this kinase and trying to integrate several decades’ worth of genetics and cell biology that laid the framework for our understanding. We trimmed a significant amount of text where we felt we could without sacrificing scientific rigor.

We also note the reviewers’ point about waiting until the Results section to introduce the *tau* allele. To satisfy this concern, we moved some of general info about the *tau* mutant to the Introduction and shortened its description in the Results section.